# 🏝 OASIS Uncovers: High-Quality T2I Models, Same Old Stereotypes

**Sepehr Dehdashtian    Gautam Sreekumar    Vishnu Naresh Boddeti**
Michigan State University
{sepehr, sreekum1, vishnu}@msu.edu

## Abstract

Images generated by text-to-image (T2I) models often exhibit visual biases and stereotypes of concepts such as culture and profession. Existing quantitative measures of stereotypes are based on statistical parity that does not align with the sociological definition of stereotypes and, therefore, incorrectly categorizes biases as stereotypes. Instead of oversimplifying stereotypes as biases, we propose a quantitative measure of stereotypes that aligns with its sociological definition. We then propose OASIS to measure the stereotypes in a generated dataset and understand their origins within T2I models. OASIS includes two scores to measure stereotypes from a generated image dataset: **(M1)** Stereotype Score to measure the distributional violation of stereotypical attributes, and **(M2)** WALS to measure spectral variance in the images along a stereotypical attribute. OASIS also includes two methods to understand the origins of stereotypes in T2I models: **(U1)** StOP to discover attributes that the T2I model internally associates with a given concept, and **(U2)** SPI to quantify the emergence of stereotypical attributes in the latent space of the T2I model during image generation. Despite the considerable progress in image fidelity, using OASIS, we uncover that newer T2I models such as FLUX.1 and SDv3 contain strong stereotypical predispositions about concepts and still generate images with widespread stereotypical attributes. Additionally, the severity of stereotypes worsens for nationalities with lower Internet footprints. *Content warning: This paper contains images with potentially offensive stereotypes.*

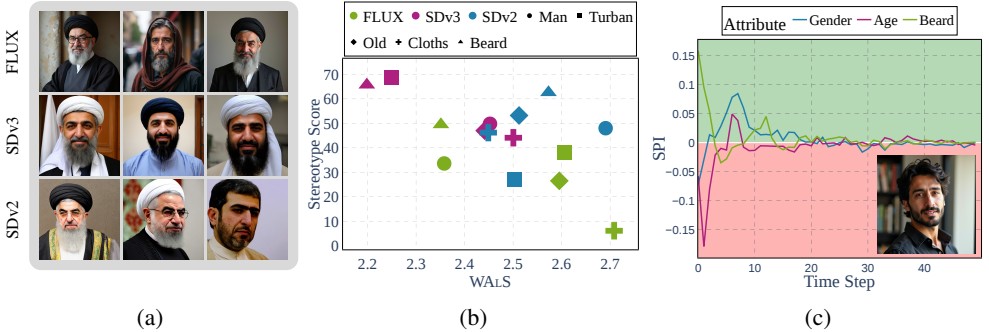

(a)    (b)    (c)

Figure 1: **Measuring Stereotypes in Text-to-Image Models.** (a) The images generated by T2I models corresponding to the prompt "*A photo of an Iranian person*" overwhelmingly contain stereotypical tropes such as *beard*, *turban*, and *religious attire* although the prompt is devoid of this information. (b) The proposed toolbox OASIS includes complementary methods for quantifying stereotypes. Stereotype Score measures the over-representation of stereotypical attributes while WALS measures the variance of images along these attributes. (c) SPI quantifies the emergence of stereotypes from the latent space of these models and helps understand the origin of stereotypes within a T2I model.

## 1 Introduction

In a sociological context, stereotypes are generalized beliefs or assumptions about a particular group of people, things, or categories (Bordalo et al., 2016). These stereotypes are widespread

in the images generated by text-to-image (T2I) models when the input textual prompts contain concepts such as culture and profession. For instance, consider the images in Fig. 1 generated by FLUX.1 (BlackForestLabs, 2024), SDv3 (Esser et al., 2024), and SDv2 (Rombach et al., 2022) for the prompt "*A photo of a/an* <nationality> *person*". There are clear portrayals of ethnic stereotypes in attributes such as *clothing*, *skin tone*, and *facial features* across different nationalities, despite no references to such attributes in the prompt. For example, the model consistently depicts an *Iranian* person as a *middle-aged* or *senior* with a *long beard*, *wearing a turban*, and dressed in *religious attire*, reinforcing harmful stereotypical representations about people with *Iranian* nationality. Besides being demographically incorrect, stereotypical biases in these models can lead to broader harm. For instance, when the biased outputs of these models are shared online, they can perpetuate damaging stereotypes about marginalized groups, further exacerbating societal polarization on issues such as beauty standards, ethnicity, and disability representation (Zhang et al., 2023; D'Incà et al., 2024; Vázquez & Garrido-Merchán, 2024).

Existing methods to detect stereotypes primarily rely on feedback from human annotators, which is both subjective and resource-intensive. It also becomes impractical in the era of the fast-paced development of generative models and changing regulations. Additionally, the feedback from human annotators may be affected by their personal and political leanings (Sap et al., 2022; Geva et al., 2019), e.g., annotation of continuous-valued attributes such as nose size and skin tone. Human annotation can also affect the users' privacy by exposing the generated images to external evaluators.

In contrast, automated methods use classifiers to detect stereotypes (Cho et al., 2023; Friedrich et al., 2023; Zhang et al., 2023), overcoming several drawbacks of human annotators. However, these methods incorrectly rely on a general bias metric, i.e., statistical parity, as a stereotype measure that fails to account for the directionality in the sociological definition of stereotypes. For example, consider a biased T2I model that generates images of predominantly female doctors. Existing works categorize this bias as a stereotype, although the generally known gender stereotype associated with the concept of *doctor* is that *all doctors are male* (Vogel, 2019).

This paper presents a new mathematical definition of stereotypes that aligns with the sociological definition. Building upon this formulation, we propose Open-set Assessment of Stereotypes in Image generative models (OASIS), a novel toolbox for quantifying stereotypes and understanding their origins in T2I models, addressing the limitations of prior studies. OASIS provides two metrics for measuring stereotypes based on the distribution and spectrum of the generated data in a feature space. OASIS comprises two additional methods to (1) discover the stereotypical attributes that a T2I model internally associates with a concept and (2) quantify the emergence of stereotypical attributes in the latent space of T2I models. Our work is an important step toward automated auditing and mitigating stereotypical content in T2I models during development and deployment.

**Contributions.**

1. We formulate the quantitative measurement of stereotypes in text-to-image (T2I) models as the over-representation of attributes that are generally associated with a concept, aligning with the sociological definition of stereotypes (Def. 1).

2. Using the new formulation, we propose OASIS, a toolbox for auditing visual stereotypes in T2I models. OASIS has two methods to measure stereotypes: **(M1)** Stereotype Score ($\Psi$) to quantify the presence of a stereotype (§ 3.1), and **(M2)** Weighted Alignment Score (WALS) to measure the spectral variety of the generated images along a given stereotypical attribute (§ 3.2).

3. In addition to methods to measure stereotypes, OASIS includes two methods for understanding the origins of stereotypes in T2I models: **(U1)** Stereotypes from Optimized Prompts (StOP) that discover stereotypical attributes that a T2I model internally associates with a given concept (§ 3.3), and **(U2)** Stereotype Propagation Index (SPI) that quantifies the emergence of stereotypical attributes during the generation steps of T2I model (§ 3.4).

4. Analyzing the scores from OASIS, we conclude that newer T2I models such as FLUX.1 and SDv3 contain strong stereotypical predispositions regarding the concepts and generate significant stereotypical biases that can limit their potential applications. Moreover, these stereotypes worsen for under-represented groups with lower Internet footprints. When these models demonstrated lower stereotypes, they did so at the cost of lower attribute variance.

## 2 STEREOTYPES IN T2I MODELS: BACKGROUND AND PROBLEM DEFINITION

Despite the growing awareness and efforts from the research community, stereotype measurement in T2I models faces challenges on two major fronts: **1) Defining a candidate set of stereotypes** for each concept is challenging due to stereotypes' subjective and evolving nature. This difficulty is further compounded by the personal biases of the candidate set designer, particularly for under-represented and lesser-known communities. Challenges regarding candidate set definition are sometimes addressed with the help of large language models (LLMs) to generate a set of potential stereotypical attributes for a given concept (Jha et al., 2023; D'Incà et al., 2024). **2) Developing analytical methods to measure known stereotypes** requires mathematical definitions and numerical quantification of visual stereotypes. Existing mathematical definitions of stereotypes do not align with the sociological definition of stereotypes. This work focuses on the analytical challenges in developing methods to measure stereotypes and understand their origins in T2I models.

**Definitions and Notations.** We use the term *concept* to refer to groups of people, things, or categories related to which stereotypes may exist, e.g., culture and profession. We denote concepts using a random variable $C$. If $C$ denotes the concept of nationality, then it takes values from $\{Iranian, American, \cdots\}$. For a given concept $C = c$, we define the set of potential stereotypical attributes as $\mathcal{A}_c \subset \mathcal{A}$, where $\mathcal{A}$ is the set of all possible attributes. Every attribute $A_i \in \mathcal{A}_c$ is a binary random variable that assumes values from $\{a_i^+, a_i^-\}$, where $a_i^+$ and $a_i^-$ indicate the presence and the absence of $A_i$, respectively. For example, if $c = Iranian$, then $\mathcal{A}_c = \{beard, religious\ symbols, hijab, \dots\}$. We denote *concepts* and *stereotypes* in different colors.

**Problem Setting.** The objective is to measure stereotypes in a T2I model $\mathcal{M}$ from the set $\mathcal{A}_c$, that purportedly exists, related to a concept $c$. For example, in Fig. 1, $c$ could correspond to *Mexican nationality* and $\mathcal{A}_c$ could include *sombrero* and *serape*. The distribution of images $I$ generated by $\mathcal{M}$ conditioned on text prompt $T(c)$ is $p_{\mathcal{M}}(I \mid T(c))$. The notation $T(c)$ indicates that the text prompt contains information about only the concept and not of any stereotype. To detect the presence of $A \in \mathcal{A}_c$, we are provided with a dataset $\mathcal{D}$ of $N$ samples generated by $\mathcal{M}$ from text prompts $T(c)$ where $\mathcal{D} := \{I_i \mid I_i \sim p_{\mathcal{M}}(I \mid T(c)), i = 1, \cdots, N\}$.

## 3 OASIS: A STEREOTYPE MEASUREMENT AND UNDERSTANDING TOOLBOX

**Motivation.** The measurement of a stereotype related to a concept is subjective without a formally defined metric. Prior works have not considered the differences between stereotypes and biases and have employed bias definitions as stereotype metrics. The dataset $\mathcal{D}$ is considered unbiased w.r.t. an attribute $A \in \mathcal{A}_c$ if

$$A \mid \mathcal{D} \sim \mathcal{U} \tag{1}$$

where $\mathcal{U}$ is uniform distribution. However, not all biases are necessarily stereotypes.

**Quantitative Measure of Stereotype.** Stereotypes are generalized beliefs or assumptions about a particular group of people, things, or categories (Bordalo et al., 2016). "Generalization" in this definition can be translated to statistical terms as exceeding the true distribution of the data for a concept $c$ in the real world. As an example, if $\mathcal{D}$ contains generated images of *doctors in the US* and the stereotype of interest $A$ is *male*, the distribution of *male* in $\mathcal{D}$ must match with its true distribution in the real world $P^*(A \mid C)$[1] i.e., $P(A = male \mid \mathcal{D}, C = Doctor) = P^*(A = male \mid C = Doctor)$. Moreover, stereotypes are directional, which means *male* having a smaller likelihood of *doctors in the US* compared to the real-world distribution is not considered a stereotype, although it is a bias. Accounting for this directionality, we say a dataset $\mathcal{D}$ contains stereotype $A$ w.r.t. $c$ if

---

**Definition 1.** *Stereotype*

$$\max(0, P(A \mid \mathcal{D}, C) - P^*(A \mid C)) \geq \zeta$$

---

where $\zeta$ is a margin for the violation from the real-world distribution. Note that this definition of stereotype differs from the definition of bias in Eq. (1). Our definition (i) compares the distribution of the generated dataset against its true societal distribution, and (ii) concerns the violation only along the direction of the attribute prone to be a stereotype.

---

[1]$P^*(A \mid C)$ can be obtained from census and online sources. For details, refer to § A.1.3.

Figure 2: **An overview of OASIS.** Given a text prompt, a set of images is generated using the T2I model $\mathcal{M}$. Simultaneously, a stereotype candidate set is created using an LLM. OASIS then performs four quantitative analyses: (M1) Stereotype Score $\Psi$ to measure stereotypes based on Def. 1, (M2) WALS to assess the spectral variance of $\mathcal{D}$ w.r.t. a stereotypical attribute, (U1) StOP to discover the stereotypical attributes that $\mathcal{M}$ associates with a given concept, and (U2) SPI to quantify the emergence of stereotypical attributes in the latent space of $\mathcal{M}$ during image generation.

**Finding Stereotype Candidates.** To find open-set stereotype candidates for a concept $c$, we follow the approach by D'Incà et al. (2024). Let $\mathcal{M}_{\text{LLM}}$ be a large language model (LLM). By providing prompt $T(c)$ and a template instruction $\mathcal{I}$[2], we have

$$\mathcal{M}_{\text{LLM}}\left(T(c), \mathcal{I}\right) = \left\{\left(A_i, d_i^+, d_i^-\right) \mid i = 1, \cdots, n_{\mathcal{A}_c}\right\} \tag{2}$$

where $d_i^+$ and $d_i^-$ are the descriptions for the presence and the absence of $A_i$, respectively. Subsequently, $\mathcal{A}_c := \left\{A_1, \ldots, A_{n_{\mathcal{A}_c}}\right\}$. For example, let $T(c)$ be "*A photo of a doctor*" and $A_i$ be *male*.[3] Here, $d_i^+$ is "*A photo of a* man" and $d_i^-$ is "*A photo of a* woman".

Based on these definitions, we propose OASIS, a toolbox to measure stereotypes in $\mathcal{M}$ from distributional and spectral perspectives and to understand the origin of these stereotypical attributes in the T2I model. Given a concept $c$, OASIS takes in as input the dataset $\mathcal{D}$ corresponding to a prompt $T(c)$, the latent space $F$ from $\mathcal{M}$ at every time step of image generation, and the candidate set of stereotypes $\mathcal{A}_c$. OASIS first extracts features $Z$ from the images using a pre-trained vision-language model (VLM) such as CLIP (Radford et al., 2021). Using these inputs, OASIS calculates the metrics we define below. Fig. 2 illustrates an overview of the proposed toolbox OASIS.

## 3.1 Stereotype Score: Measuring Stereotypes in T2I Models

Following Def. 1, stereotype score ($\Psi$) of $A \in \mathcal{A}_c$ for a given dataset $\mathcal{D}$ and concept $c$ is defined as

$$\Psi\left(A \mid \mathcal{D}, C\right) := \max(0, P(A \mid \mathcal{D}, C) - P^*(A \mid C)) \tag{3}$$

where $P^*(A \mid C)$ is the real-world density of $A$ in concept $c$. Using Bayes' rule, $P(A \mid \mathcal{D}, C)$ is,

$$P(A \mid \mathcal{D}, C) \propto \prod_{i=0}^{N} P(A \mid I_i, C)$$

$$P(A = a^+ \mid \mathcal{D}, C) = \frac{\prod_{i=0}^{N} P(A = a^+ \mid I_i, C)}{\sum_{a'} \prod_{i=0}^{N} P(A = a' \mid I_i, C)} \tag{4}$$

We obtain $P(A \mid I_i, C)$ by means of attribute classifiers. Instead of training attribute-specific classifiers, a zero-shot predictor such as CLIP (Radford et al., 2021) can be used, where $P(A \mid I_i, C)$ is obtained using a softmax over cosine similarity scores of image features and text descriptions for $a^+$ and $a^-$. However, these cosine similarity scores are often numerically close (Liang et al., 2022),

---

[2]Refer to § A.1.1 for more details on the template instruction.

[3]The number of categories for gender is restricted by the annotations of the existing datasets.

requiring an additional temperature parameter to obtain accurate probability measures. Therefore, in such cases, we estimate $P(A \mid I_i, C)$ as

$$P(A = a^+ \mid I_i, C) = \mathbb{1}\left(\langle Z_I, Z_{a^+}\rangle_{\cos} > \langle Z_I, Z_{a^-}\rangle_{\cos}\right) \tag{5}$$

where $\langle x, y \rangle_{\cos}$ is the cosine similarity between $x$ and $y$, $\mathbb{1}$ is the indicator function, and $Z_I$, $Z_{a^+}$, and $Z_{a^-}$ are features of image, $d^+$, and $d^-$ from Eq. (2), respectively.

## 3.2 $\underline{\text{WALS}}$: Measuring Spectral Variety along a Stereotype

**Motivation.** Since $\Psi$ measures stereotypes from a distributional perspective, it is possible for a dataset $\mathcal{D}$ to appear free of stereotypes at the cost of reduced variance along the stereotypical attribute. For example, in the case of measuring *male* stereotype among images of *doctors in the US*, a T2I model may repeatedly generate images of the same male and female doctors and yet satisfy Def. 1. Moreover, it is challenging to measure variety through human inspection due to its subjective nature, and therefore, a quantitative method to inspect variance is beneficial. To encapsulate these requirements, we propose a metric named Weighted Alignment Score (WALS) that measures the spectral alignment of the data $\mathcal{D}$ with a given attribute $A$.

**Method.** To quantify the changes in a given stereotypical attribute $A$ across images generated by a T2I model, WALS involves two steps: **1) Estimating the structure of data $\mathcal{D}$** through the singular value decomposition of the CLIP image features $\mathcal{E}_I(\mathcal{D})$ i.e., $\mathcal{E}_I(\mathcal{D}) = U\Sigma V^T$ where $\mathcal{E}_I$ is the image encoder of the CLIP model, **2) Finding the direction of change in $A$**, denoted by $\delta A$, using one of the following two approaches: (i) estimating $\delta A$ as the difference between the text embeddings of a pair of positive and negative descriptions, $d^+$ and $d^-$,

$$\delta A = \mathcal{E}_T\left(d^+\right) - \mathcal{E}_T\left(d^-\right), \tag{6}$$

where $\mathcal{E}_T$ is the text encoder of the CLIP model, or (ii) estimating $\delta A$ as the direction of maximum change along $A$ in the image embedding space of a set of $A$-aware images corresponding to positive ($a^+$) and negative ($a^-$) categories of $A$, using supervised principal component analysis (Barshan et al., 2011). Detailed descriptions and derivations can be found in § A.2. These approaches make different assumptions, and one of these can be chosen based on the available information and the availability of computational resources. The first approach assumes alignment between text and image embeddings in the CLIP model, and $\delta A$ is more accurate when the embeddings of these modalities are more aligned. The second approach estimates $\delta A$ accurately at the cost of increased computation due to generating two $A$-aware image sets and calculating the kernel matrices. Moreover, the first approach captures linear dependency, while the second one can be adopted for both linear and non-linear dependencies. We use the former approach in our experiments. Using the two components explained above, WALS measures the data variance along $\delta A$ in the feature space, as

$$\text{WALS}(A) := \frac{\sum_{i=1}^{k} \sigma_i \cdot \delta A^T u_i}{\sum_{j=1}^{k} \sigma_j} \tag{7}$$

where $\sigma_i$ is $i^{\text{th}}$ singular value of $\mathcal{D}$, and $u_i$ is the associated singular vector.

## 3.3 $\underline{\text{STOP}}$: Discovering Internally Associated Stereotypical Attributes

**Motivation.** Stereotypes might occur due to T2I models internally associating a concept $c$ with stereotypical attributes. This means that the prompts with these attributes can equivalently generate images corresponding to $c$. However, these attributes may not be present in $\mathcal{A}_c$. Therefore, qualitative methods are devised to discover these open-set attributes, which we refer to as $\mathcal{M}$-*attributes*.

**Method.** Since the distribution of stereotypical attributes is not uniform within $\mathcal{D}$, we have to find $\mathcal{M}$-attributes for individual clusters of images that share common stereotypes. Given an image dataset corresponding to concept $c$, we use spectral clustering (Von Luxburg, 2007) on CLIP features extracted from these images and visually identify clusters that share stereotypes. To discover $\mathcal{M}$-attributes for a given cluster with prominent stereotypes, we design a sequence optimization problem, following ZeroCap (Tewel et al., 2022). The solution to this optimization problem is a sequence that maximizes its mean CLIP score with the images in the chosen cluster. Formally, with a cluster

$\mathcal{D}' = \{I_1, \ldots, I_n \mid 1 \le i \le n\}$ containing $n$ images, the objective is

$$s^* = \arg\max_s \frac{1}{n} \sum_{i=1}^{n} \langle \mathcal{E}_T(s), \mathcal{E}_I(I_i) \rangle_{\cos} \tag{8}$$

where $\mathcal{E}_I$ and $\mathcal{E}_T$ are image and text encoders from CLIP. Following ZeroCap, $s$ is produced by an LLM[4] that is conditioned on the starting sequence "*This is a photo of* ". The subsequent optimization problem reduces to iteratively finding a 2-token sequence that maximizes the mean CLIP score in Eq. (8) using beam search. Since a single prompt $s^*$ may not contain diverse stereotypical attributes, we output the top-$K$ prompts in the final iterative step of the optimization in Eq. (8).

### 3.4 SPI: UNDERSTANDING THE EMERGENCE OF STEREOTYPES IN T2I MODELS

**Motivation.** In addition to measuring stereotypes from generated images, it is important to quantify the aggregation of stereotypical attributes during image generation to design successful mitigation strategies. To that end, we propose stereotype propagation index (SPI) to quantify the addition of stereotypical attributes in the latent space of $\mathcal{M}$ at each time step of image generation.

**Method.** In the flow-based models such as SDv3, the latent in each inference step is updated as $x_{t+1} = x_t + v_\Theta(x_t, t, \epsilon_t)$ where $x_t$ and $x_{t+1}$ are the latent representation in the current and next step, respectively, $v_\Theta(.)$ is the velocity of $x_t$ for time step $t$, and $\epsilon_t = \epsilon_\Theta(x_t, t, c_p)$ is the noise predicted in time step $t$ for latent $x_t$ by the noise predictor $\epsilon_\Theta$, where $c_p$ is the conditioning text prompt. The velocity decides the attributes of the generated image based on the provided text prompt. Our goal is to measure the amount of a stereotypical attribute added during each step of image generation, which requires knowing the direction of change in the attribute ($\delta A$) in the latent space of the T2I model.

To find $\delta A$ in the latent space of the T2I model, we first predict two $A$-aware noises that correspond to positive $d^+$ and negative $d^-$ descriptions of $A$ as

$$\epsilon_t^+ = \epsilon_\Theta(x_t, t, d^+) \qquad \epsilon_t^- = \epsilon_\Theta(x_t, t, d^-). \tag{9}$$

Using these predicted noises, we find the velocities that model could have in this step if the text prompt was $A$-aware, i.e., $v_\Theta(x_t, t, \epsilon_t^+)$ and $v_\Theta(x_t, t, \epsilon_t^-)$. Here, the direction of change in the attribute can be calculated as

$$\delta A = v_\Theta(x_t, t, \epsilon_t^+) - v_\Theta(x_t, t, \epsilon_t^-). \tag{10}$$

We define SPI as the cosine similarity between the velocity at time step $t$ and the direction of change in the given attribute $A$:

$$\text{SPI}(A, t) := \left\langle v_\Theta(x_t^i, t, \epsilon_t), \delta A \right\rangle_{\cos} \tag{11}$$

A positive SPI means the stereotypical attribute is being added to the image in time step $t$, and a negative SPI means that the image is losing the stereotypical attribute $A$.

## 4 WHAT DOES OASIS UNCOVER ABOUT STEREOTYPES IN T2I MODELS?

We apply OASIS on three open-weight T2I models – SDv2 (Rombach et al., 2022), SDv3 (Esser et al., 2024), and FLUX.1[dev] (BlackForestLabs, 2024). In the first step, as illustrated in Fig. 2, we generate a dataset of 2000 images of people from each of the *nationalities* and with each T2I model. In the next step, for each *nationality*, a candidate set for stereotypes and their descriptions is generated according to Eq. (2). We used ChatGPT o1-preview (OpenAI, 2024b) and ChatGPT 4o (OpenAI, 2024a) as $\mathcal{M}_{\text{LLM}}$ in Eq. (2). Implementation details are mentioned in § A.1.

### 4.1 LOWER, YET SIGNIFICANT STEREOTYPES IN NEWER T2I MODELS

We use CLIP ViT-G-14 from OpenCLIP (Ilharco et al., 2021) trained on LAION2B (Schuhmann et al., 2022) to estimate $P(A \mid \mathcal{D}, C)$. Tab. 1 compares the T2I models in terms of their stereotype scores defined in § 3.1 from the images generated by these models corresponding to three *nationalities* – *Iranian*, *Indian*, and *Mexican*. Although the fidelity of the generated images has improved dramatically

---

[4]We use Llama 3.1 (Dubey et al., 2024)

Table 1: **Stereotype Score.** Comparison of three T2I models, SDv2, SDv3, and FLUX.1 on stereotype score in three *nationalities*. $P^*(A \mid C)$ is the true density of the attribute obtained from real-world statistics (details provided in § A.1.3), $P(A \mid \mathcal{D}, C)$ is the density of the attribute in the generated dataset, and $\Psi(A \mid \mathcal{D}, C)$ is the stereotype score. All values are in %.

| | Stereotype Candidate | $P^*(A \mid C)$ | SDv2 | | SDv3 | | FLUX.1[dev] | |
| | | | $P(A \mid \mathcal{D}, C)$ | $\Psi(A \mid \mathcal{D}, C)$ | $P(A \mid \mathcal{D}, C)$ | $\Psi(A \mid \mathcal{D}, C)$ | $P(A \mid \mathcal{D}, C)$ | $\Psi(A \mid \mathcal{D}, C)$ |
|---|---|---|---|---|---|---|---|---|
| *Iranian* | *Man* | 50 | 98 | 48 | 99.8 | 49.8 | 83.6 | 33.6 |
| | *Wearing Turban* | 0.2 | 27.3 | 27.1 | 69 | 68.8 | 38.2 | 38 |
| | *Old* | 40 | 93.2 | 53.2 | 87 | 47 | 66.5 | 26.5 |
| | *Traditional Cloths* | 50 | 96.2 | 46.2 | 94.1 | 44.1 | 56.1 | 6.1 |
| | *Beard* | 34 | 96.6 | 62.6 | 99.7 | 65.7 | 83.5 | 49.5 |
| *Indian* | *Man* | 51 | 78.5 | 27.5 | 78.1 | 27.1 | 31.6 | 0 |
| | *Turban* | 2 | 2.2 | 0.2 | 0.9 | 0 | 0.1 | 0 |
| | *Mustache* | 25 | 17.7 | 0 | 12.4 | 0 | 25.9 | 0.9 |
| | *Tilak/Bindi* | 50 | 61.7 | 11.7 | 59.3 | 9.3 | 86.7 | 36.7 |
| | *VibrantColorCloths* | 50 | 41.5 | 0 | 58.3 | 8.3 | 53.8 | 3.8 |
| *Mexican* | *Man* | 48 | 95.1 | 47.1 | 85 | 37 | 50.1 | 2.1 |
| | *Hat* | 50 | 77.3 | 22.3 | 49.2 | 0 | 94.4 | 44.4 |
| | *Sombrero* | 50 | 56.6 | 6.6 | 17.6 | 0 | 58.6 | 8.6 |
| | *Mustache* | 25 | 77.8 | 52.8 | 34.1 | 9.1 | 84.7 | 59.7 |
| | *Embroidered Clothing* | 50 | 82.6 | 32.6 | 45.9 | 0 | 94.2 | 44.2 |

from SDv2 to SDv3 and FLUX.1, our results demonstrate that stereotype scores of newer models are generally lower than those of the older ones. However, in some cases, there are exceptions. As an example, when generating images of *Mexican person*, FLUX.1 depicts 84.7% of the faces with *mustache* while SDv2 and SDv3 generate 77.8% and 34.1% faces with *mustache*, respectively. In high-level attributes such as gender, FLUX.1 has lower stereotype scores than other models. For example, in the case of *Iranian*, FLUX.1 depicts 83.6% of images as *man*. But in comparison, 98% and 99.8% of the images generated by SDv2 and SDv3, respectively, depict *man*.

> **Remark.** Existing bias definitions are not applicable for some attributes studied in Tab. 1. E.g., a T2I model needs to depict 50% of the images of *Iranian* with *turban* to be unbiased according to Eq. (1), which incorrectly represents *Iranian people* among whom only 0.2% wear *turban*.

Previous works have noted the gender imbalance in the generated images for certain professions such as *doctors* and *teachers* (D'Incà et al., 2024). We observe a similar trend in the newer T2I models as shown in Tab. 2. However, SDv3 has a lower gender imbalance compared to SDv2 for *doctor*. We hypothesize that this is due to the data balancing methods taken to ensure unbiased gender representation in the images of *doctor* following the scrutiny it has faced. However, the imbal-

Table 2: $P(A = man \mid C, \mathcal{D})$ for $C = doctor$, $C = Iranian\ doctor$, and *Indian Doctor*.

| Model | *Doctor* | *Indian Doctor* | *Iranian Doctor* |
|---|---|---|---|
| SDv2 | 93 | 97 (**+4**) | 98 (**+5**) |
| SDv3 | 78 | 98 (**+20**) | 100 (**+22**) |
| FLUX.1 | 93 | 100 (**+7**) | 100 (**+7**) |

ance worsens when a *nationality* is added to the profession (e.g., *Iranian doctor*). This example demonstrates that stereotype mitigation through data balancing is insufficient against intersectional stereotypes as it is infeasible to collect data samples corresponding to every possible combination.

## 4.2 IMAGES OF UNDER-REPRESENTED NATIONALITIES CONTAIN MORE STEREOTYPES

T2I models are often trained on image-caption pairs that are scraped from the Internet. Therefore, their training data may be biased by the Internet footprint of various nationalities. To investigate the impact of a nationality's Internet footprint on stereotypes in T2I models, we compare the stereotype scores of generated images from various nationalities against their corresponding number of Internet users. We consider generated images corresponding to *Indian*, *Mexican*, and *Iranian* nationalities, which have populations of 881.3 million, 96.8 million, and 78.1 million, respectively (WorldPopulationR, 2024). Fig. 3 presents the stereotype scores across different attributes for each country and model. The results indicate that the maximum and the average stereotype scores for a nationality decrease as the number of Internet users increases. These findings suggest that the stereotypes in T2I models may be exacerbated for under-represented nationalities when trained on image-caption pairs from the Internet.

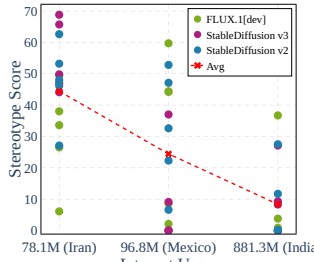

Figure 3: Comparing stereotype scores for nationalities against the number of Internet users shows that stereotypes are higher for under-represented nationalities.

### 4.3 EFFECTIVE T2I MODEL COMPARISON REQUIRES BOTH STEREOTYPE SCORE AND WALS

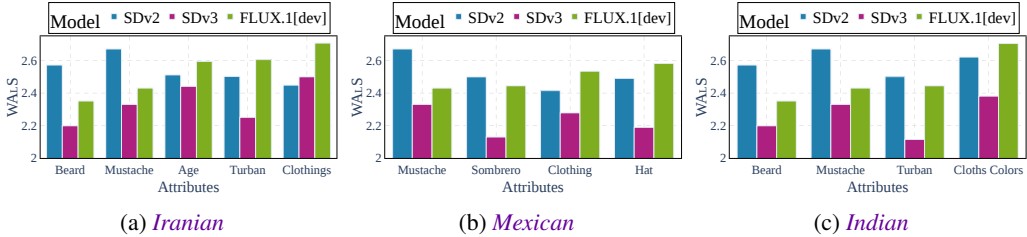

Figure 4: **WALS**: Comparison of SDv2, SDv3, and FLUX.1 on spectral variance in the generated images across different attributes, calculated for *Iranian*, *Mexican*, and *Indian* nationalities.

Fig. 4 compares the WALS for T2I models on three nationalities. A higher WALS($A$) indicates more variance in the images along the attribute $A$. We observe that FLUX.1 generates images that show a higher variety in clothing items such as *hats* and *turbans*, but have a lower variance regarding facial attributes such as *beard* and *mustache* across all three *nationalities*. In contrast, images generated by SDv2 show a higher variance on *beard* and *mustache* than on *clothing-related* attributes.

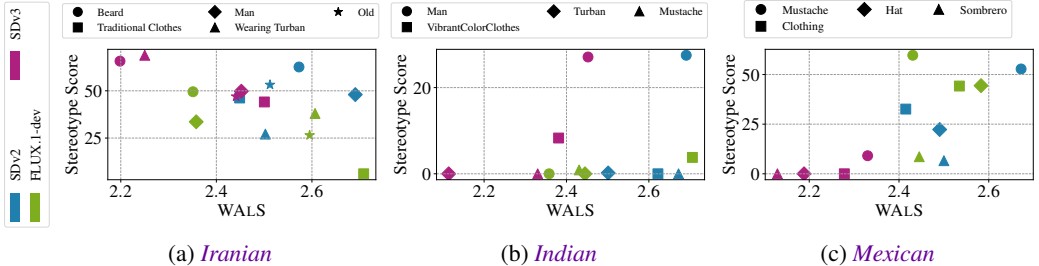

Figure 5: Comparison of T2I models based on stereotype scores and WALS on three *nationalities*. Different colors show different T2I models and the shapes of the markers denote the *attributes*.

As mentioned in § 3.2, stereotype score and WALS are complementary measures of stereotypes. We can compare the models jointly on these scores to verify if models demonstrate lower stereotypes at the cost of lower variety. Fig. 5 plots stereotype score and WALS for various T2I models and attributes for each *nationality*. An ideal T2I model must have a low stereotype score and a high WALS and therefore must appear towards the bottom-right corner of these plots. We observe that some models have lower stereotypes while having lower attribute variance. For instance, images from SDv3 tend to have lower WALS across all *nationalities*, although they succeed in reducing stereotypes in some attributes. These observations highlight the importance of employing both distributional (Stereotype Score) and spectral (WALS) metrics together to compare the T2I models.

### 4.4 T2I MODELS INTERNALLY ASSOCIATE CONCEPTS WITH STEREOTYPES

We use StOP to discover the internal associations that the T2I model $\mathcal{M}$ makes with a given concept $c$. In Tab. 3, we show $\mathcal{M}$-attributes in FLUX.1 discovered using StOP for three concepts: *Iranian*, *Mexican*, and *American*. We obtain clusters of images using spectral clustering on the CLIP features of aligned faces and manually identify those with shared stereotypes. The average of the faces in the clusters are shown in the second column. The attributes that we expect StOP to discover can be visually identified from these averaged images. For example, the average of the cluster corresponding to *Iranian* shows an *old man* wearing a *turban* and sporting a *long beard*, characteristic of the Islamic religious leaders in Iran. Therefore, the expected $\mathcal{M}$-attributes include religious terminology. In the third column, we show some of the optimized prompts that StOP produces. The optimized prompts contain Unicode characters in vernacular languages. The optimized prompts corresponding to *Iranian* images include religious terms such as "*Imam*" and "*Sheikh*". Similarly, the optimized prompts for *Mexican* images include "*brero*" (short for *sombrero*). In the last column, we input one of these prompts to FLUX.1 to visually inspect the resulting images. Unsurprisingly, the images generated from these optimized prompts are visually similar to those generated from prompts containing only *nationality*. For example, *the US national flag* can be seen as a blurred background in the cluster average and is also present in the samples generated from optimized prompts for *American person*.

Table 3: **StOP** first identifies image clusters for each concept using spectral clustering. The averages of the images from these clusters are shown in the second column. StOP finds the captions shown in the third column by solving the optimization problem in Eq. (8). These captions contain stereotypical attributes such as "*Imam*" and "*brero*". The fourth column shows the images generated using these optimized captions. Unsurprisingly, these images contain insignia of the corresponding culture.

| Culture | Cluster average | Optimized prompts | Samples from highlighted prompt |
|---------|-----------------|-------------------|--------------------------------|
| *Iranian* | | "This is a photo of \u093f\u092e Imam" "This is a photo of \u0935 reb" "This is a photo of \u093f\u0935 Sheikh" | |
| *Mexican* | | "This is a photo of brero mayor" "This is a photo of bero Garcia" "This is a photo of brero pastor" | |
| *American* | | "This is a photo of EO Democrat" "This is a photo of :border counselor" "This is a photo of :border ambassador" | |

### 4.5 STEREOTYPICAL ATTRIBUTES EMERGE IN THE EARLY STEPS OF IMAGE GENERATION

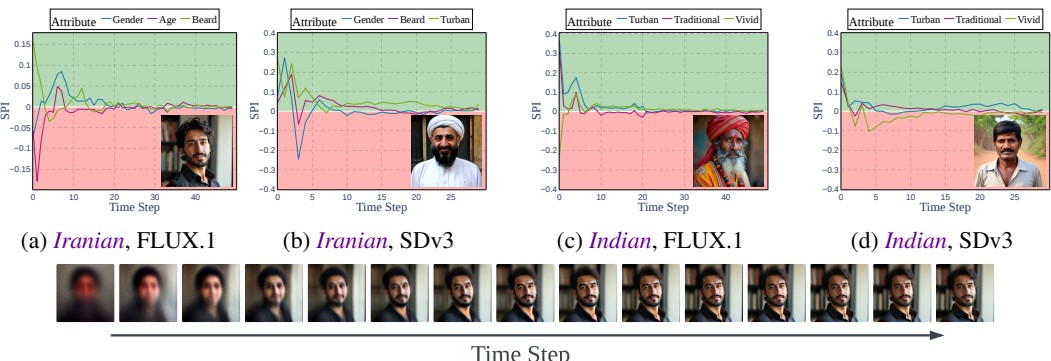

(a) *Iranian*, FLUX.1  (b) *Iranian*, SDv3  (c) *Indian*, FLUX.1  (d) *Indian*, SDv3

Time Step

Figure 6: **SPI** tracks the change in attributes in the image generation processes of FLUX.1 and SDv3. We observe that these attributes are affected during the early time steps of image generation.

We quantify the emergence of stereotypical attributes during image generation in FLUX.1 and SDv3 for image prompts of the form "*A photo of an <nationality> person*" using SPI. For a given stereotype, we first obtain positive and negative descriptions corresponding to it. For example, for the attribute *age*, *old* and *young* were used in $d^+$ and $d^-$ in Eq. (2), respectively. SPI is then calculated as the cosine similarity between $\delta A$ and velocity of the latent $x_t$ at time step $t$ as shown in Eq. (11). We plot $SPI(A, t)$ for all time steps during the generation of four images from *Iranian* and *Indian* nationalities in Fig. 6. We observe that a relatively high amount of information on attributes such as *beard*, *traditional cloths*, and *sombrero* is added to the images at the first step of generation in FLUX.1 indicating that the model readily associates these attributes with the concept. Specifically, we observe that the stereotypical attributes arise during the earlier time steps of generation in both *Iranian* and *Indian* images. In the example of *Iranian* person in Fig. 6a, we observe that *age* and *beard* attributes form in the image within the first 3 time steps and *gender* attribute emerges at time step 7. After time step 20, the changes in these attributes are negligible. Likewise, for *Indian* nationality, stereotypical attributes undergo little change after time step 20. Additional results for other nationalities and the average SPI over 100 samples are provided in § A.3 and Fig. 9, respectively.

### 4.6 T2I MODELS HAVE STEREOTYPICAL PREDISPOSITIONS ABOUT CONCEPTS

In § 4.5, we noted that stereotypical attributes aggregate in the early steps of image generation. A question that naturally follows this observation is: *are T2I models predisposed to*

*generate stereotypical images for a given concept?* This can be answered by considering the velocity $v_\Theta(x_t, t, \epsilon_t)$ of the early time steps since they guide towards the mean of the data distribution (Kynkäänniemi et al., 2024). This enables us to identify the stereotypical predispositions qualitatively. For each time step $t$, we estimate the final time step image $\hat{x}_T$ based on velocity $v_\Theta(x_t, t, \epsilon_t)$ as $\hat{x}_T = x_t + v_\Theta(x_t, t, \epsilon_t)(T - t)$, as illustrated in Fig. 7a. Fig. 7b shows these images for three samples corresponding to *Iranian person*. The images generated using the velocity at time $t = 0$ appear to be of a person with *turban* and *beard*, even when the final generated images lack these attributes. Conflating with our observations from § 4.4 and § 4.5, we conclude that T2I models associate stereotypical attributes with seemingly innocuous prompts. Additional results are provided in § A.4.

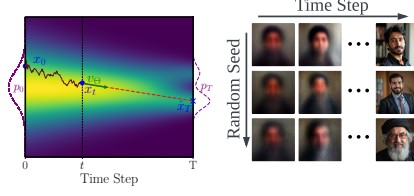

(a) Estimation     (b) *Iranian person*

Figure 7: Stereotypical predisposition in T2I models for *Iranian person*.

## 5   RELATED WORK

Many studies have shown that deep learning models tend to learn and, at times, amplify the biases present in their datasets (Bolukbasi et al., 2016; Buolamwini & Gebru, 2018; Luccioni & Viviano, 2021; Blodgett et al., 2021; Agarwal et al., 2021; Aka et al., 2021; Sadeghi et al., 2022; Birhane et al., 2023; 2024; Chuang et al., 2023; Dehdashtian et al., 2024b;a;c; Phan et al., 2024), and T2I models are no exception. Most of the existing work about stereotypes in T2I models has focused on gender and ethnic biases in the generated images. Some studies have shown that prompts play a significant role in the bias generated by T2I models (Bansal et al., 2022; Zhang et al., 2023; Seshadri et al., 2024). Seemingly neutral prompts lead to geographical biases favoring Western nations such as the US and Germany, leading to lighter skin tones and Western norms in the images (Bianchi et al., 2023; Naik & Nushi, 2023), while prompts containing certain cultural and gender terms sometimes generate NSFW images, reflecting the biases in the training datasets (Birhane et al., 2021; Ungless et al., 2023; Schramowski et al., 2023). Luccioni et al. (2024) measured distributional biases in professions w.r.t a closed set of genders and ethnicities.

Unlike these works, we use an open set of stereotypes obtained from an LLM, following (D'Incà et al., 2024). Although these studies have achieved breadth in terms of sources for stereotypes, they have primarily used statistical parity as the definition of stereotype. For example, (Jha et al., 2024) uses "stereotype tendency" defined as the ratio of the likelihood of a stereotype appearing in a group to that of it appearing in the general population, ignoring the directionality of stereotypes. In contrast, we measure stereotypes following their true sociological definition. We additionally provide insights into the origins of the stereotypical attributes in T2I models.

## 6   CONCLUDING REMARKS

This paper proposed OASIS to measure and understand the origin of stereotypes in T2I models based on a quantitative measure that aligns with the sociological definition of stereotype. OASIS includes: (M1) Stereotype Score (§ 3.1) to measure the directional violation of the true stereotypical attribute distribution in the T2I model, (M2) WALS (§ 3.2) to measure the spectral variety of the generated images along the stereotypical attributes, (U1) StOP (§ 3.3) to discover the stereotypical attributes that the T2I model internally associates with a concept, and (U2) SPI (§ 3.4) to measure the emergence of stereotypical attributes during image generation from the latent space. Despite the considerable progress in the image fidelity of T2I models, using OASIS, we conclude that the newer models such as FLUX.1 and SDv3 have strong stereotypical predispositions about concepts and still struggle to avoid stereotypical attributes in the generated images.

**Recommendations.** OASIS unveils the extent of stereotypes in T2I models. However, commonly pursued solutions for correcting biases in generative models such as data balancing are not suitable for resolving stereotypes due to the sheer number of concepts that could potentially have stereotypes. Additionally, concepts such as *nationalities* worsen stereotypes in unrelated concepts such as *doctors* as observed in Tab. 2. It is infeasible to collect data samples at the intersection of multiple concepts. Therefore, training-time mitigation and post hoc correction techniques that are tailored to remove stereotypes in T2I models must be developed (Phan et al., 2024; Zhang et al., 2023; Friedrich et al., 2023; Parihar et al., 2024). Our observations also underscore the need for increased participation of under-represented communities in the development of large generative models.

**Acknowledgements:** This work was supported in part by the National Science Foundation (award #2147116) and the Office of Naval Research (award #N00014-23-1-2417). Any opinions, findings, and conclusions or recommendations expressed in this material are those of the authors and do not necessarily reflect the views of NSF or ONR.

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

# A    APPENDIX

In our main paper, we proposed OASIS for quantifying stereotypes and understanding their origins in T2I models. Here, we provide some additional analysis to support our main results. The appendix section is structured as follows:

1. Implementation Details in § A.1

2. Finding $\delta A$ Using $A$-Aware Generated Images in § A.2

3. More Results on SPI and average SPI in § A.3

4. More Results on T2I models' Stereotypical Predispositions in § A.4

5. Limitations in § A.5

6. Ethical Considerations in § A.6

7. Importance of Detecting Stereotypes in § A.7

8. Qualitative Descriptions of the Generated Datasets in § A.8

## A.1    IMPLEMENTATION DETAILS

In this section we provide all the details that are employed in the experiment section in order to obtain the mentioned results.

### A.1.1    LLM TEMPLATE INSTRUCTION

To generate an open-set candidate for stereotypes corresponding to a given *concept*, we follow the approach provided in (D'Incà et al., 2024). The template instruction $\mathcal{I}$ that was used in Eq. (2) is as follows:

> I am using a text-to-image model to generate facial images of individuals from various nationalities. These images depict faces along with a surrounding margin that may include headwear or clothing visible above the chest. I need a candidate set of at least 15 potential stereotypes to monitor for in the generated images. I will provide the prompts used to generate the images, and I would like you to identify this candidate set. Additionally, I plan to check the presence or absence of these attributes in the images using a CLIP model. For each attribute, I need at least two sentences: one that positively describes the existence of the attribute and another that describes its absence.

An example of the output produced by ChatGPT o1-preview is,

> Certainly! Based on your prompt, "*A photo of a Tunisian person,*" here is a list of potential attributes to look for in the generated images. These attributes are associated with cultural elements and common depictions of Tunisian people. For each attribute, I've provided a positive and a negative sentence that can be used with a CLIP model for zero-shot classification.
>
> 1. Wearing a Traditional Chechia (Hat)
>    - Positive: "The person is wearing a traditional Tunisian chechia."
>    - Negative: "The person is not wearing any headwear."
> 2. Traditional Clothing
>    - Positive: "The person is wearing traditional Tunisian clothing."
>    - Negative: "The person is wearing modern Western clothing."
> 3. Wearing a Djebba (for men)
>    - Positive: "The man is wearing a traditional djebba."
>    - Negative: "The man is not wearing a djebba."
>
>    ⋮
>
> 15. Wearing a Sefsari (for women)
>     - Positive: "The woman is wearing a traditional sefsari."
>     - Negative: "The woman is not wearing a sefsari."

### A.1.2 IMAGE GENERATION PROMPTS

To generate images from FLUX.1 and SDv3, we used the prompt template "*A photo of <nationality> person*", where *<nationality>* could take values like *American*, *Indian*, etc. For SDv2, we observed that the images could be of lower quality or contain the torso of the person. Therefore, for each images, we sampled a prompt randomly from the following array of prompts: "*A photo of <nationality> person*", "*A picture of <nationality> person*", "*A portrait photo of <nationality> person*", "*A front profile photo of <nationality> person*".

### A.1.3 OBTAINING TRUE DISTRIBUTIONS

Stereotype score is measured as the violation of the true underlying distribution of an attribute given a concept, denoted by $P^*(A \mid C)$, in the generated images. One could obtain $P^*(A \mid C)$ from official census data and online statistcs. For example, Brotherton & Etzel (2023) provides various demographic details about doctors in the US such as ethnicity and gender in various specializations. For attributes where it is difficult to obtain precise statistics, e.g., traditional clothing, we consider their presence a choice and assign a 50% chance for their presence. For example, for *mustache* for people from *Mexican* nationality, we calculate $P^*(mustache \mid Mexican) = 0.5 \times P^*(male \mid Mexican) \approx 0.255$.

### A.2 FINDING $\delta A$ USING $A$-AWARE GENERATED IMAGES

As mentioned in § 3.2, to find the direction of change in $A$, we propose two approaches: (i) using text embeddings of a pair of positive and negative descriptions, $d^+$ and $d^-$, and (ii) using $A$-aware generated images. The first approach is explained in § 3.2 and in this section, we explain how to find $\delta A$ using $A$-aware generated images in both linear and non-linear cases.

### A.2.1 LINEAR $\delta A$

As mentioned in § 3.2, using $A$-aware generated images to find $\delta A$ can be more precise than using text embeddings. As an example, for finding the direction of change in *male* for images corresponding to "*A photo of an* Iranian person", two sets of images using prompts "*A photo of a* man" and "*A photo of a* woman" are created. The set of CLIP features of these images are denoted by $Z_A = \{z_i\}_{i=1}^m$ and their corresponding labels of $A$ as $Y_A = \{y_i\}_{i=1}^m$. Then we find an orthogonal transformation matrix $\Gamma$ that maps $Z_A$ to a subspace that maximizes the variance of the labeled data using supervised principal

component analysis (Barshan et al., 2011) that maximizes the dependency between the mapped data $\Gamma^\top Z_A$ and $Y_A$. Hilbert-Schmidt Independence Criterion (HSIC) (Gretton et al., 2005) is employed as the dependence metric where its empirical version is defined as $\text{HSIC}^{\text{emp}} = \text{Tr}\{HK_{ZZ}HK_Y\}$, where $H$ is the centering matrix, $K_Y$ is a kernel matrix of $Y$, and $K_{ZZ}$ is a kernel matrix of the mapped data. When using a linear kernel, it becomes $K_{ZZ} = Z^\top \Gamma\Gamma^\top Z$. Therefore, $\Gamma$ can be calculated by solving the following optimization

$$\arg\max_{\Gamma} \text{Tr}\{\Gamma^\top \boxed{ZHK_{YY}HZ^\top} \Gamma\}, \tag{12}$$

$$\text{subject to} \quad \Gamma^\top\Gamma = I \tag{13}$$

This optimization has a closed-form solution, and the columns of the optimal $\Gamma$ are the eigenvectors of $M := \boxed{ZHK_{YY}HZ^\top}$ corresponding to the $d$ largest eigenvalues where $d$ is the dimensionality of the subspace (Lütkepohl, 1997). Here, since we only need a direction vector, we choose the eigenvector $\hat{v}_1$ associated with the largest eigenvalue of $M$.

$$\delta A = \Gamma = \hat{v}_1. \tag{14}$$

To capture the non-linear relations of the attribute, a non-linear kernel can be used to calculate $K_Y$ and $K_{ZZ}$. The closed-form solution for the non-linear case is provided in § A.2.2.

### A.2.2 NON-LINEAR $\delta A$

If we are interested in finding non-linear relations between the images in order to find direction of change in an attribute, a non-linear version of the formulation mentioned in the previous subsection can be used. In this approach, similar to the linear case, we generate two sets of images associated with positive ($a^+$) and negative ($a^-$) categories of $A$. The set of CLIP features of these images are denoted by $Z_A = \{z_i\}_{i=1}^m$ and their corresponding labels of $A$ as $Y_A = \{y_i\}_{i=1}^m$. Then we find an orthogonal transformation matrix $\Gamma$ that maps kernelized $Z_A$ to a subspace that maximizes the variance of the labeled data using supervised principal component analysis (Barshan et al., 2011) that maximizes the dependency between the mapped data $\Gamma^\top K_{ZZ}$ and $Y_A$.

Hilbert-Schmidt Independence Criterion (HSIC) (Gretton et al., 2005) is employed as the dependence metric where its empirical version is defined as $\text{HSIC}^{\text{emp}} = \text{Tr}\{HK_{ZZ}HK_Y\}$, where $H$ is the centering matrix, $K_Y$ is a kernel of $Y$, and $K_{ZZ}$ is the kernelized $Z_A$ using a similarity measure of the mapped data. $\Gamma$ can be calculated by solving the following optimization

$$\arg\max_{\Gamma} \text{Tr}\{\Gamma^\top \boxed{K_{ZZ}HK_{YY}HK_{ZZ}^\top} \Gamma\}, $$

$$\text{subject to} \quad \Gamma^\top\Gamma = I \tag{15}$$

This optimization has a closed-form solution and the optimal solution for $\Gamma$ are the eigenvectors of $M := \boxed{K_{ZZ}HK_{YY}HK_{ZZ}^\top}$ corresponding to the $d$ largest eigenvalues where $d$ is the dimensionality of the subspace (Lütkepohl, 1997). Here, since we only need a direction vector, we choose the eigenvector $\hat{v}_1$ associated with the largest eigenvalue of $M$.

$$\delta A = \Gamma = \hat{v}_1. \tag{16}$$

### A.3 MORE RESULTS ON SPI

**More Sample-Wise Results.** More samples for SPI on *Mexican person* and *Tunisian person* are illustrated in Fig. 8.

**Average SPI.** The average SPI in 100 images generated by SDv3 corresponding to "*A photo of an Iranian person*" is demonstrated in Fig. 9. As illustrated, the T2I model adds a high amount of information on attributes such as *turban* in the earlier steps. This confirms our earlier conclusions from sample-wise SPI in Fig. 6b. Additionally, we note that the variance for SPI is small in the time step $T = 0$, suggesting the stereotypical predispositions noted in § 4.6. However, in the next few time steps, we see a slightly larger variance that indicates that these models tend to correct the stereotypical attributes added in the former time steps.

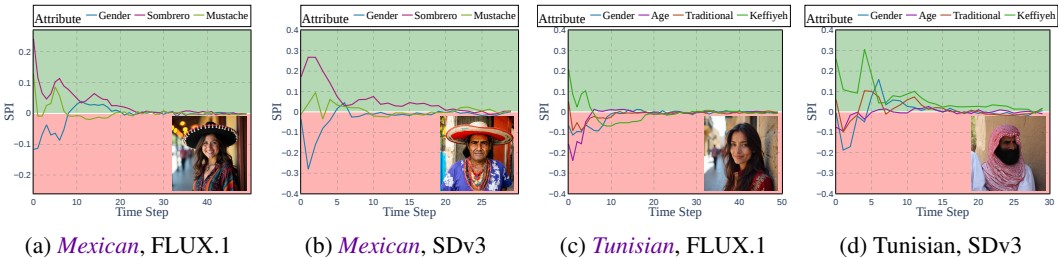

Figure 8: Additional samples of SPI plots of generated images by FLUX.1 and SDv3 for *Mexican* and *Tunisian* nationalities. A positive value for SPI($A, t$) means that $a^+$ is added to the image at time step $t$. Similarly, a negative SPI means that the image is moving toward $a^-$.

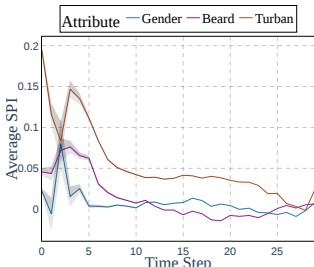

Figure 9: Average SPI in 100 samples for *Iranian person* generated by SDv3.

## A.4 MORE RESULTS ON T2I MODELS' STEREOTYPICAL PREDISPOSITIONS

In this section, we provide additional results that show that T2I models are predisposed to create stereotypical images for various *nationalities*. In Fig. 10, we show additional results for FLUX.1 on *Iranian*, *Indian*, and *Mexican* nationalities. Similar to our observations in § 4.6, we note that the images generated from the velocity at $t = 0$ for *Iranian* person contain stereotypical attributes such as *beard* and *turban*. Likewise, for images of *Indian* personality, we observe *vibrant clothing* (e.g., orange veil). In the images of *Mexican* person, we can see a faded *sombrero* in the early images. Moreover, the attributes that appear in the early stages of image generation are absent in the final generated image, indicating that these stereotypical attributes arise due to their intrinsic association with the concept.

## A.5 LIMITATIONS

**Obtaining $P^*(A \mid C)$.** Access to $P^*(A \mid C)$ is a crucial component for any stereotype measuring method. As mentioned in § A.1.3, $P^*(A \mid C)$ is obtained from census data and online sources when they are available. We note that reliable sources may not be available for every attribute and changes in survey methods can affect the results. However, for most stereotype evaluation and mitigation applications, reliable data can be found from government and survey agencies.

**Use of CLIP.** As mentioned in § 3.1, we obtain $P(A \mid I_i, C)$ using attribute classifiers. Instead of training attribute-specific classifiers, a zero-shot predictor like CLIP (Radford et al., 2021) can be utilized. However, some attributes may be unfamiliar to the model, resulting in lower accuracy in detecting them within the images. Additionally, these models may be biased in terms of concepts such as ethnicity. With advancements in zero-shot prediction models and the introduction of more accurate versions, newer models can seamlessly replace the existing ones in OASIS, thanks to its modular design.

For a small dataset of *doctors* that is used in Tab. 2, we evaluate the performance of the CLIP model in predicting the *gender*. As mentioned earlier, for each T2I model, we generated 100 images of *doctors* and manually labeled their genders. The accuracy of the CLIP model in predicting *gender* is demonstrated in Tab. 4. The results suggest that on this small dataset, the CLIP model can predict *gender* almost as accurately as human annotators.

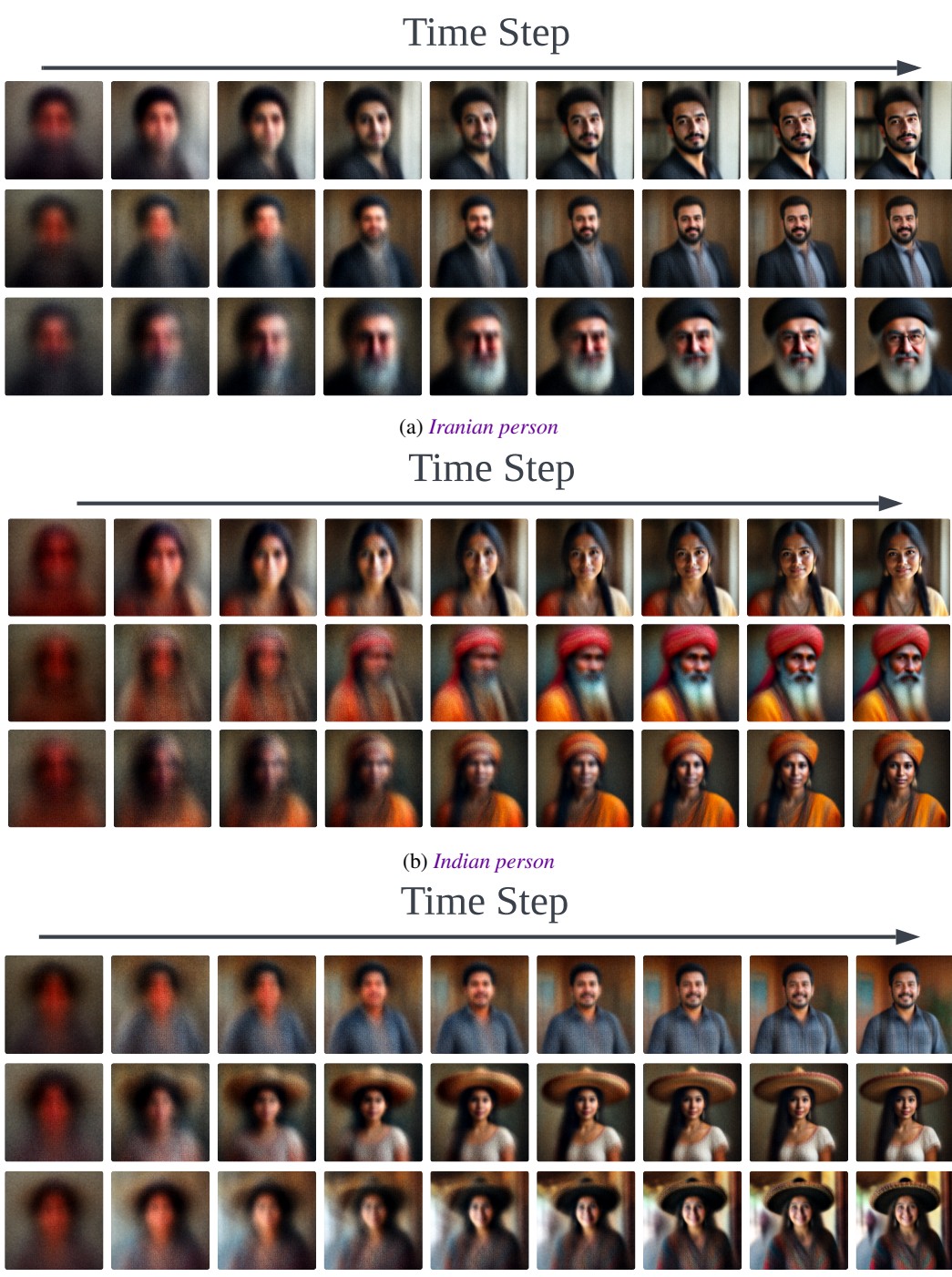

(a) *Iranian person*

(b) *Indian person*

(c) *Mexican person*

Figure 10: First 9 steps of image generation in FLUX.1 model for three nationalities: (a) *Iranian person*, (b) *Indian person*, and (c) *Mexican person*

Additionally, we evaluate the performance of the employed CLIP model on CelebA (Liu et al., 2015) that contains more than 200,000 face images of celebrities annotated with 40 binary attributes. Since we primarily used CLIP to predict attributes such as *beard* and *hat*, we evaluate the model on similar attributes (i.e., *having beard*, *man*, *wearing a hat*, *having a mustache*). The accuracy in predicting each attribute is reported in Tab. 5. The results demonstrate that the CLIP model can

Table 4: Performance of the CLIP model on predicting *gender* in the generated *doctors* dataset.

| Model | Accuracy |
|---|---|
| SDv2 | 100% |
| SDv3 | 99% |
| FLUX.1 | 99% |

predict the attribute with an acceptable accuracy. As we noted earlier, although the CLIP model may not accurately predict certain general attributes, our results indicate that the CLIP model is suitable for predicting the attributes that we considered in this work.

Table 5: Performance of the CLIP model on predicting four attributes in CelebA dataset.

| Attribute | Accuracy |
|---|---|
| *having beard* | 83.06% |
| *gender* | 99.38% |
| *wearing a hat* | 96.14% |
| *having a mustache* | 94.77% |

The above-mentioned experiments, show the effectiveness of using a CLIP model in automating the classification of the images. However, the accuracy of the model is not 100% which indicates that by newer vision-language model with higher accuracy compared to the CLIP model that is employed in this paper, should be replaced in the OASIS.

## A.6 ETHICAL CONSIDERATIONS

In this work, we measure stereotypes and explore their origins in images generated by publicly available T2I models. We find that these models pose ethical concerns by reinforcing social stereotypes and potentially offending some users. By introducing OASIS, we aim to highlight these issues and move the community one step closer to mitigating stereotypes in generative models.

## A.7 IMPORTANCE OF DETECTING STEREOTYPES IN GENERATIVE MODELS

Visual content produced by generative models inadvertently perpetuates stereotypes about various ethnicities, cultures, nationalities, and professions (Ananya, 2024). Such images and videos are shared on online social media accounts such as X and Reddit, and this can reinforce stereotypical notions about certain social groups. This content could also influence public perception of marginalized communities and could undermine ongoing efforts to integrate them into mainstream society. For example, it has been noted that generated images of women from certain ethnicities tend to be sexualized (Lamensch, 2023). Additionally, the adoption of these generative models by various companies and institutions may have unforeseen consequences. For example, Nicoletti & Bass (2023) states that using generative AI to develop suspect sketches could lead to wrongful convictions.

## A.8 QUALITATIVE DESCRIPTIONS OF THE GENERATED DATASETS

We evaluated OASIS on the images corresponding to various nationalities generated by different T2I models. In this section, we give a qualitative description of the generated images. A few *randomly* selected representative samples from each culture and T2I model are shown in Fig. 11.

**FLUX.1.** The images produced by FLUX.1 are of high quality and look realistic. However, some stereotypes can be qualitatively observed from the images in Fig. 11a. For example, some images of *American* people contain the *American flag*. Images of *Indian* people tend to show vibrant colored clothing. Most of the generated images of *Iranian* people are of *men* and most of them wear *turban*. Among the images of *Mexican* people, *sombrero* is the most common stereotypical element. We additionally note a general superficial diversity among the samples. For example, the images in each row were generated with the same random seeds. We can observe that the backgrounds in these photos

are sometimes repeated. For instance, compare the first columns of *Indian* and *Mexican* samples. Additionally, there is a clear disparity in the backgrounds across nationalities. The backgrounds for *American* and *Iranian* images are more often indoors than for *Indian* and *Mexican*. Images of *American* and *Iranian* people also more often contain images of officials compared to *Indian* and *Mexican* images. Interestingly, Donald Trump's image appeared when prompted to generate images of *American* person.

**SDv3.** Fig. 11b shows the samples generated by SDv3. Although the generated images are of high fidelity, unlike FLUX.1, they lack variety in background and poses. Surprisingly, images of *American* people are relatively free of stereotypes and show ethnic diversity. However, the face images of *Indian* people look very similar and contain elements such as *tilak/bindi*. The diversity drops further in the images of *Iranian* people. All the randomly selected samples contained images of *men* wearing *turban* and *religious attire*. Among the images of *Mexican* people, *sombreros* were present but in fewer proportions compared to the images generated by FLUX.1.

**SDv2.** Some representative samples generated by SDv2 are shown in Fig. 11c. Among the considered T2I models, SDv2 produced images with the least photorealism, with some displaying distorted facial expressions. However, these images generally contain diverse facial attributes such as hairstyle. Images of *American* people are of higher quality compared to other nationalities, although they include black & white portraits. We note the lack of ethnic diversity among these images compared to those in SDv3 and FLUX.1. Although identity diversity is lower for images of *Indian* people compared to FLUX.1 and SDv3, we also observe fewer stereotypical attributes. Similar to SDv3, the images of *Iranian* people generated by SDv2 are primarily of *men*, mostly donning *turban*. Stereotypes such as *sombrero* and *colorful clothing* are present in the images of *Mexican* people. Among all the T2I models that we considered, SDv2 seems to have the least gender diversity across all nationalities.

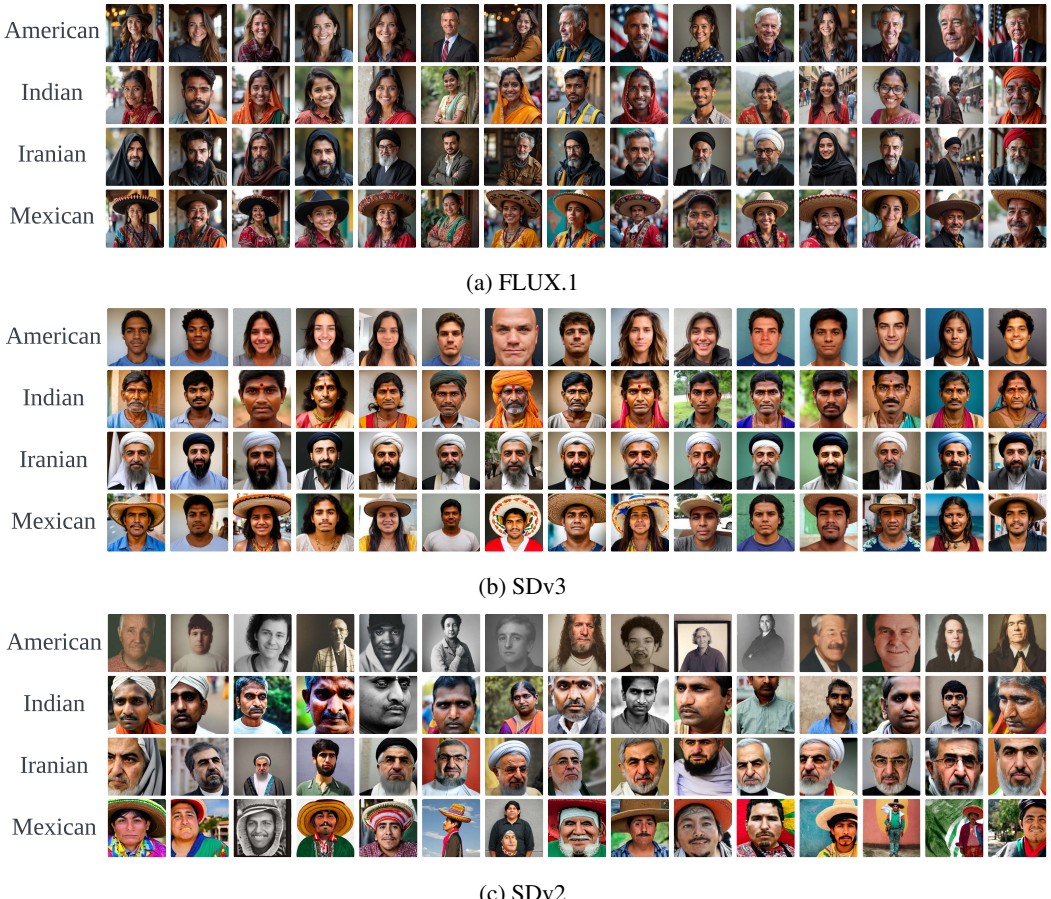

Figure 11: A few *randomly* selected representative samples from each culture and T2I model.

