# OpenReview forum: "OASIS Uncovers: High-Quality T2I Models, Same Old Stereotypes"
_ICLR.cc/2025/Conference — ICLR 2025 Spotlight_

### Official Review · Reviewer_QudX · 2024-10-29

**Soundness:** 2
**Presentation:** 3
**Contribution:** 2
**Rating:** 6
**Confidence:** 4

**Summary:**

This paper aims to measure and understand stereotypes in Text-to-Image models.

* Measurement: it introduces metrics to quantify the presence of stereotypes in generated images by comparing them to real-world distributions, and also assess diversity along stereotypical attributes.

* Track Origins:  The author attempts to identify hidden associations that T2I model makes with certain concepts and trace how stereotypical attributes emerge over the generation process

**Strengths:**

1. The paper tackles the significant problem of stereotypes in T2I models, which has important implications for ethical AI use and social impact.
2. The paper is clearly written and organized. The visual presentations are good.
3. It proposes StOP and SPI that try to understand the origin of the stereotypes, which may offer further insights.

**Weaknesses:**

1. The authors emphasize that their method is innovative in focusing specifically on stereotypes rather than just bias in the abstract. However, this distinction is not fully explained in the main content, and the methodology still seems quite similar to bias detection to me.

2. The way they identify stereotypes is by prompting LLMs to generate stereotypical attributes for a given concept, but this method is already common in other papers. For example, [1] prompts LLMs to generate stereotypical / anti-stereotypical sentences for certain concepts and then identifies the stereotypical / anti-stereotypical attributes from those sentences.

3. Although the paper includes many formulas, I found that the core idea is actually quite simple, and the formulas only make it harder to follow. If I understand correctly, the main process is: identify a concept, use LLMs to generate stereotypical attributes, generate images, measure the appearance of these attributes, and compare to real-world distributions & similarities across groups. This could be communicated in just a few sentences, which would make the approach clearer.

4. I am concerned with the idea of comparing to real-world distributions. For example, men are dominant in many professions even today. In society, an educated person can be aware of stereotypes to some extent, but telling the model that it’s okay to learn patterns such as “men are more likely to be doctors” because it reflects real-world distributions doesn’t seem fully convincing to me.

[1] chatgpt based data augmentation for improved parameter-efficient debiasing of llms

**Questions:**

I would appreciate it if the authors addressed the questions mentioned in the weaknesses section, and I will raise the score if their responses are convincing.

---

> ### Author Response · Authors · 2024-11-17
> **Response to Reviewer QudX**
>
> Thank you for your feedback. Below are our responses to individual questions and concerns.
>
> > **W1** The authors emphasize that their method is innovative in focusing specifically on stereotypes rather than just bias in the abstract. However, this distinction is not fully explained in the main content, and the methodology still seems quite similar to bias detection to me.
> >
>
> Please refer to the common response “Difference between existing bias definitions and our stereotype definition”.
>
> > **W2** The way they identify stereotypes is by prompting LLMs to generate stereotypical attributes for a given concept, but this method is already common in other papers. For example, [1] prompts LLMs to generate stereotypical / anti-stereotypical sentences for certain concepts and then identifies the stereotypical / anti-stereotypical attributes from those sentences.
> >
>
> As mentioned in L110-120, we categorize the challenges in stereotype measurement in T2I models on two major fronts: 1) Defining a candidate set of stereotypes, and 2) Developing analytical methods to measure the stereotypes. As mentioned there, the first challenge is addressed by the prior works with the help of LLMs to generate a set of potential stereotypes attributes for a given concept (Jha et al., 2023; D’Incà et al., 2024). In this paper, we address the second challenge and use the methods proposed for identifying a candidate set of stereotypes from prior studies. This point is demonstrated in the overview of OASIS in Figure 2.
>
> > **W3** Although the paper includes many formulas, I found that the core idea is actually quite simple, and the formulas only make it harder to follow. If I understand correctly, the main process is: identify a concept, use LLMs to generate stereotypical attributes, generate images, measure the appearance of these attributes, and compare to real-world distributions & similarities across groups. This could be communicated in just a few sentences, which would make the approach clearer.
> >
>
> The summary that is provided by the reviewer is correct only for the Stereotype Score (Section 3.1) which is one of the four proposed metrics. The other metrics in OASIS measure spectral variance along stereotypical attributes, discover the stereotypical attributes internally associated with the given concept, and quantify the emergence of attributes during the image generation process.
>
> Even for the Stereotype Score that the reviewer mentioned, mathematical notations are essential to bring clarity and avoid the probable ambiguity in the text descriptions. To make the descriptions clear and easy to follow, we minimized the mathematical equations and moved most of them to the appendix sections.
>
> > **W4** I am concerned with the idea of comparing to real-world distributions. For example, men are dominant in many professions even today. In society, an educated person can be aware of stereotypes to some extent, but telling the model that it’s okay to learn patterns such as “men are more likely to be doctors” because it reflects real-world distributions doesn’t seem fully convincing to me.
> >
>
> In response to the reviewer's concern, we would like to clarify that the Stereotype Score in OASIS measures the presence of stereotypes in the generated images based on the definition of stereotype. If the goal is to have a stereotype-free model, the score should be zero. However, if additional considerations, such as achieving gender balance in generation, are important, other metrics, such as general statistical parity, can be used to evaluate and guide those aspects.
>
> To clarify this point using detailed examples of doctors we have added a section in our common response (“On the ethics of using census data to measure stereotype”). We kindly refer the reviewer to that section for further details.
>
> If our responses addressed your initial comments, please consider raising the score.

---

> > ### Comment · Reviewer_QudX · 2024-11-23
> >
> > Thanks for the prompt response, I have updated my score accordingly

---

> > > ### Author Response · Authors · 2024-11-23
> > >
> > > Thank you for updating your score.
> > > Given that the updated score remains below the acceptance threshold, we would like to understand if there are any outstanding concerns or areas where our responses were unclear. If there are specific points that require further clarification or additional revisions, we would be happy to address them.

---

> > > > ### Comment · Reviewer_QudX · 2024-12-03
> > > >
> > > > Thank you for your response. I have reviewed all the points you raised during the rebuttals and have no further questions. I have updated my score accordingly.

---

### Official Review · Reviewer_AMAa · 2024-11-04

**Soundness:** 3
**Presentation:** 4
**Contribution:** 3
**Rating:** 6
**Confidence:** 4

**Summary:**

The authors present OASIS, a set of 4 metrics/methods to analyze the presence of stereotypes in T2I models, how those stereotypes form, and the connections between them. The authors use 3 popular models for this analysis. In the paper, OASIS shows the performance of these models on various prompts (e.g. "Show me a picture of a Mexican person") and how certain stereotypes are strongly associated with the prompt, at rates greater than those stereotypes in the real world. The paper shows weaknesses in models in their attempts to prevent stereotypes such as revealing stereotypes when prompts or combined or in how they produce low-variance along the stereotype's dimension. The paper also shows how certain stereotypes are clustered together and at what stage in the image generation process these models form (which is almost always in the early stages).

In all, OASIS provides a useful toolkit for understanding stereotypes in T2I models and highlights the importance for further work in this very important area.

**Strengths:**

The paper was very well-written, and it was quite easy to follow. I greatly appreciated how the terminology was constantly linked to concrete examples. The general formatting was also intuitive.

The paper did a good job of thinking about the broader context of this work, thinking a bit more critically about what phrases like biases actually mean and what we should think about when studying these models. Also incorporating real-world census data was a neat idea.

I liked the results about how the biases are being generated early in the image-generation process.

The formulation for the stereotype score was quite intuitive and seems easy enough to compute at scale when provided with accurate descriptors from these LLMs.

**Weaknesses:**

I feel that overall, there was a lack of bench-marking for the methods presented in this paper beyond providing a few visual examples (which are still useful!). There were a few parts of the paper where I felt this significantly

1. Computing the stereotype score: How accurate is the probability computation? The intuition of the metrics' formulation is clear, but there is no validation that the score provides accurate results in practice. As the paper mentions, CLIP is a biased VLM on its own, so its not fair to assume that this score will work perfectly. A smaller dataset demonstrating the accuracy of the score could be quite useful.

2. Spectral variety along a stereotype. There was a nice description of each method for measuring $\delta A$ in the appendix, but its not clear which one is used in the paper and which one is more accurate in practice.

3. The biases of models like CLIP are well-documented (e.g. Birhana, Prabhu, & Kahembwe, 2021). If CLIP is being used extensively to identify additional biases, I think it makes sense to add more validation about how said biases may or may not be affecting the performance of OASIS.

4. I understand the SPI claims about how certain stereotypes are intrinsically associated with a concept. However, the paper only provides a handful of examples of where this is occurring. I would appreciate a larger-scale analysis that includes something like 100 contexts generated in some way (i.e. top 100 most common ethnicities), and the average step at which each stereotype attribute occurs. This would be much more convincing than what could be cherry-picked examples.

I think that the breadth of the experiments/validation were also a bit limited. I understand that gaining access to real-world data for a ground truth distribution is not always feasible, but consistently focusing on broad nationalities feels like it lacks nuance.

Another broad critique is that certain figures were unclear. I'll bring up those missing details in the following sections

**Questions:**

Is this definition of stereotypes correct? Does it not implicitly push the stereotypes in society that cause imbalances to exist? Relying on census data to define an expected rate of a given attribute is perpetuating the stereotypes that exist in today's society. For example, using the example provided in the paper, if there are less women than male doctors in the world, it doesn't necessarily mean that the model is creating a 'stereotype' by generating an equal number of woman and male doctor images.

For experimental details, what is each point in Figure 3? What does each point correspond to in terms of attribute?

In Figure 5, there is a claim that "We observe that some models have lower stereotypes at the cost of lower attribute variance". This suggests a trade-off can exist between these two, but there is not much correlation observed for any model across nationalities. I agree that both metrics are important to measure, but it would be great to get more clarification on if there indeed is a trade-off that occurs often.

I would appreciate any additional discussion about the accuracy of the metrics as well and if they work as intended.

---

> ### Author Response · Authors · 2024-11-17
> **Response to Reviewer AMAa (Part 1)**
>
> We thank the reviewer for their constructive and detailed feedback. Below are our responses to individual concerns and questions.
>
> > **W1** Computing the stereotype score: How accurate is the probability computation? The intuition of the metrics' formulation is clear, but there is no validation that the score provides accurate results in practice. As the paper mentions, CLIP is a biased VLM on its own, so its not fair to assume that this score will work perfectly. A smaller dataset demonstrating the accuracy of the score could be quite useful.
> >
>
>  As per the reviewer’s suggestion, we evaluate the accuracy of the CLIP model in predicting gender on a small image dataset of doctors containing 100 images generated from the prompt “A photo of a doctor” in App. A.5, Tab. 4. Moreover, we include the accuracy of the chosen CLIP model on the CelebA dataset for attributes similar to those evaluated in the paper, presented in App. A.5, Tab. 5. The results of these evaluations indicate that the CLIP model demonstrates high accuracy for the attributes analyzed in this paper.
>
> > **W2** Spectral variety along a stereotype. There was a nice description of each method for measuring δA in the appendix, but its not clear which one is used in the paper and which one is more accurate in practice.
> >
>
> Thank you for noting that. Although we have discussed each method’s pros and cons in detail, we used the text-based approach (method (i) in L237) for the experiments due to its efficiency. We have added this in Sec. 3.2.
>
> > **W3** The biases of models like CLIP are well-documented (e.g. Birhana, Prabhu, & Kahembwe, 2021). If CLIP is being used extensively to identify additional biases, I think it makes sense to add more validation about how said biases may or may not be affecting the performance of OASIS.
> >
>
> As mentioned in the limitation and related work sections, we are aware of the biases in CLIP’s predictions. These biases can potentially affect the accuracy of the attribute prediction. However, we are not using CLIP to identify the biases; instead, we use it as a classifier to count the number of occurrences of a specific attribute in the images instead of using human labeling or training an attribute-specific classifier from scratch for each attribute. And as we show in App. A.5, Tabs. 4 and 5, CLIP is highly accurate in predicting attributes such as hat and beard that we primarily considered in this paper.
>
> Additionally, we emphasize that the potential biases in CLIP do not invalidate our comparisons of different T2I models, as the same CLIP model is used for all comparisons. All metrics are calculated for the same set of candidate attributes, ensuring a fair comparison between the T2I models.
>
> As newer and more accurate VLMs become available, the credibility of OASIS’s results and conclusions will only improve. Specifically, the contributions of our work—namely, the metrics for quantifying stereotypes, measuring spectral variance along attributes, discovering internally associated stereotypical attributes, and analyzing their emergence in T2I models—are independent of the quality of the classifier used.
>
> > **W4** I understand the SPI claims about how certain stereotypes are intrinsically associated with a concept. However, the paper only provides a handful of examples of where this is occurring. I would appreciate a larger-scale analysis that includes something like 100 contexts generated in some way (i.e. top 100 most common ethnicities), and the average step at which each stereotype attribute occurs. This would be much more convincing than what could be cherry-picked examples.
> >
>
> SPI was proposed to quantify the emergence of stereotypical attributes during image generation in a **per-sample manner**, unlike other metrics of OASIS that quantified the stereotypes from the image distribution. We considered eight samples (including the appendix) from different nationalities and noted that specific stereotypical attributes emerged early in the image-generation process. This observation is consistent across nationalities and random seeds, even in cases where the noted stereotypes were absent in the final generated image. For example, in Fig. 6d, SPI showed additional information about turban and vivid colored clothing, although the final image did not have that. Sec. 4.6 corroborated this conclusion.
>
> To further confirm this conclusion, following the reviewer’s suggestion, we have averaged SPI across 100 images of Iranian people and included those results in the App. A.3, Fig. 9. We hope these strengthened conclusions support the use of SPI to know which image-generation step to intervene in (and by how much) to control the stereotypical attribute in the final image.

---

> > ### Author Response · Authors · 2024-11-17
> > **Response to Reviewer AMAa (Part 2)**
> >
> > > I think that the breadth of the experiments/validation were also a bit limited. I understand that gaining access to real-world data for a ground truth distribution is not always feasible, but consistently focusing on broad nationalities feels like it lacks nuance.
> > >
> >
> > The primary utility of OASIS is measuring and understanding open-set stereotypes, irrespective of their concept. We chose to evaluate stereotypes associated with nationalities since every nationality is associated with wildly different stereotypes, each with a significant historical context. For example, “turban” and “hijab” for an Iranian person vs. “sombrero” and “mustache” for a Mexican person. Measuring and analyzing stereotypes in various nationalities is also an underexplored topic in existing works.
> >
> > Existing works primarily focus on bias detection in images of various professions. However, most professions share stereotypes such as gender, age, and race, and this is, therefore, not a strong use case for our proposed approach. As a result, we chose to measure the stereotypes about nationalities in depth. Additionally, we studied the intersectional stereotypes between nationalities and professions in Table 2 of the paper. It shows that OASIS is applicable beyond the measurement of stereotypes in nationalities.
> >
> > > **Q1** Is this definition of stereotypes correct? Does it not implicitly push the stereotypes in society that cause imbalances to exist? Relying on census data to define an expected rate of a given attribute is perpetuating the stereotypes that exist in today's society. For example, using the example provided in the paper, if there are less women than male doctors in the world, it doesn't necessarily mean that the model is creating a 'stereotype' by generating an equal number of woman and male doctor images.
> > >
> >
> > In the provided example by the reviewer, the Stereotype Score will not classify the model as stereotypical due to the directional nature of the stereotype definition we proposed. According to this definition, the model exhibits a gender stereotype for "doctors" only if the proportion of male doctors exceeds their proportion in the census data. In all other scenarios—whether there are more female doctors than male or an equal number of male and female doctors—the Stereotype Score will be zero.
> >
> > For more detailed examples, please refer to the common response “On the ethics of using census data to measure stereotype”.
> >
> > > **Q2** For experimental details, what is each point in Figure 3? What does each point correspond to in terms of attribute?
> > >
> >
> > Each point corresponds to an attribute in one of the three T2I models (SDv2, SDv3, and FLUX). The attributes and the stereotype scores are mentioned in Table 1. In this figure, we wanted to show how much the internet footprint of nationalities can affect the extent of stereotypes in T2I models. The results suggest that both the maximum and the average stereotype scores for the attributes decrease with the increasing internet footprint of people from various nationalities.
> >
> > > **Q3** In Figure 5, there is a claim that "We observe that some models have lower stereotypes at the cost of lower attribute variance". This suggests a trade-off can exist between these two, but there is not much correlation observed for any model across nationalities. I agree that both metrics are important to measure, but it would be great to get more clarification on if there indeed is a trade-off that occurs often.
> > >
> >
> > Sorry for the confusion, and thank you for noting that. We wanted to convey that these two metrics complement each other and both must be considered when evaluating a T2I model. We do not mean that a trade-off exists or that having a high value for one leads to a lower value for the other. We have changed the text and title of Sec. 4.3 to convey our desired message.
> >
> > If our responses addressed your initial comments, please consider raising the score.

---

> > > ### Comment · Reviewer_AMAa · 2024-11-24
> > >
> > > Thank you for your detailed responses - I have updated my score, but I have a few additional (small) clarifying questions.
> > >
> > > I believe that there are still some limitations in this approach such as the reliance on a candidate set of stereotypes (and it being an entirely binary choice) as well as census data. A perhaps useful addendum to this paper could be a discussion of approaches to help lift these limitations.
> > >
> > > Can you also provide some clarification for section 4.5 and there being larger SPI in earlier time-steps. I am convinced this is a repeatable pattern now, but what does a larger SPI in earlier steps mean about the nature of these models? How should one interpret the significance of SPI being larger in earlier time-steps than in later ones?

---

> > > > ### Author Response · Authors · 2024-11-25
> > > >
> > > > > Thank you for your detailed responses - I have updated my score, but I have a few additional (small) clarifying questions. I believe that there are still some limitations in this approach such as the reliance on a candidate set of stereotypes (and it being an entirely binary choice) as well as census data. A perhaps useful addendum to this paper could be a discussion of approaches to help lift these limitations.
> > > > >
> > > >
> > > > Thank you for revisiting your score and for providing additional suggestions.
> > > >
> > > > As outlined in the paper, OASIS provides metrics and analysis for a given set of generated images from a T2I model and a corresponding candidate set of stereotypes. The process of identifying this candidate set is not a contribution of our work and is independent of OASIS, as depicted in Figure 2. In this paper, we adopted a commonly-used approach involving LLMs to derive candidate stereotypes, following prior work in multimodal models (Adila et al., 2024) and generative models (D’Inc et al., 2024). However, OASIS is not restricted to this approach and can work with alternative methods, such as curating the candidate set through domain experts.
> > > >
> > > > To incorporate your suggestion, we will enhance the discussion in the Limitation section (A.5) to explicitly outline these points and mention some possible alternatives for identifying candidate stereotypes beyond LLMs.
> > > >
> > > > Adila, D., Shin, C., Cai, L., & Sala, F. (2024). Zero-Shot Robustification of Zero-Shot Models. *The Twelfth International Conference on Learning Representations (ICLR)*.
> > > >
> > > > > Can you also provide some clarification for section 4.5 and there being larger SPI in earlier time-steps. I am convinced this is a repeatable pattern now, but what does a larger SPI in earlier steps mean about the nature of these models? How should one interpret the significance of SPI being larger in earlier time-steps than in later ones?
> > > > >
> > > >
> > > > A larger SPI for an attribute indicates that the T2I model is adding a high amount of information related to that attribute to the image, while a lower SPI suggests minimal changes in the attribute.
> > > >
> > > > For example, in Fig. 6a, the model begins by introducing a large amount of information for the “having a beard” attribute. In later steps, this information stabilizes, resulting in fewer changes to the beard attribute. This aligns with the estimated images for each time step shown in Fig. 6 (bottom row), where the beard’s shape remains unchanged after a certain point.
> > > >
> > > > One application of SPI that we find particularly valuable is its use in methods for controlling the generation process. SPI provides guidance on which steps and to what extent the normal flow of the model should be modified.
> > > >
> > > > For example, in Olmos et al. (2024), the step number and intervention strength are treated as hyperparameters, requiring extensive trial-and-error tuning for each attribute and concept. By using SPI as a guide, this process can be automated, as it identifies the specific steps where attribute information needs adjustment.
> > > >
> > > > Olmos, Carolina López, et al. "Latent Directions: A Simple Pathway to Bias Mitigation in Generative AI." CVPR ReGenAI (2024).

---

> > > > > ### Comment · Reviewer_AMAa · 2024-11-25
> > > > >
> > > > > Thank you for your responses! I understand your points and do not have any further suggestions/clarifications that I would like to convey.

---

### Official Review · Reviewer_Hz7R · 2024-11-05

**Soundness:** 3
**Presentation:** 4
**Contribution:** 3
**Rating:** 8
**Confidence:** 3

**Summary:**

This paper introduces OASIS, a toolbox for auditing visual stereotypes in text-to-image (T2I) models. OASIS offers novel methods for measuring stereotypes and analyzing their origins within T2I models. OASIS stereotype scores reveal that newer T2I models produce significant stereotypical biases, potentially limiting their applicability.

**Strengths:**

- The paper is well-written, clear, and easy to follow.
- The paper addresses a complex and socially significant problem by evaluating biases and stereotypes in T2I models. Additionally, it provides new methods to trace the origins of these stereotypes in T2I models, offering valuable insights to help mitigate these biases.
- The findings of this work highlight the need for greater inclusion and representation of underrepresented communities, particularly those from regions with limited internet presence, in the development of T2I models.

**Weaknesses:**

- This work focuses exclusively on generated images based on nationality, which raises several concerns. For instance, in racially diverse countries like the United States, how can an image indicate that someone is a U.S. citizen without relying on stereotypes and social biases? Additionally, in regions where people from neighboring countries may share similar facial features or cultures (such as North and South Korea), how useful can OASIS be in distinguishing these cases?
- How can OASIS be useful beyond nationality domains? It is not clear in the paper how OASIS can be extended to evaluate other biases in T2I models, such as gender biases, racism, etc.
- Overall, this is a solid piece of work; however, a deeper discussion of its limitations and the specific types of stereotypes for which OASIS may or may not be effective would be beneficial.

**Questions:**

Please, refer to the weaknesses section.

**Details Of Ethics Concerns:**

This work labels people's images to their nationalities, which might sound offensive or unethical to some. Further review from the ethics experts would be necessary.

---

> ### Author Response · Authors · 2024-11-17
> **Response to Reviewer Hz7R**
>
> Thank you for your review and the encouraging feedback.
>
> > **W1** This work focuses exclusively on generated images based on nationality, which raises several concerns. For instance, in racially diverse countries like the United States, how can an image indicate that someone is a U.S. citizen without relying on stereotypes and social biases? Additionally, in regions where people from neighboring countries may share similar facial features or cultures (such as North and South Korea), how useful can OASIS be in distinguishing these cases?
> >
>
> In racially diverse countries like the US, a stereotype-free image can contain people from various ethnicities. Our objective is not to measure how closely the generated images of the people agree with the nationality in the prompt. OASIS measures the over-representation of attributes such as the American flag in these images. In the case of nationalities such as North Korean and South Korean, the prompts in Fig. 2 will differ (e.g., “A photo of a North Korean person” vs. “A photo of a South Korean person”), leading to distinct candidate sets of stereotype attributes. For example, the generated candidate set for stereotypes of a North Korean person includes Military Uniforms, Serious Facial Expression, Traditional Hanbok, and Kim Jong-un-style Haircut, while the candidate set for a South Korean person includes K-pop Style, Bright Appearance, Urban Background, and Trendy Hairstyle. OASIS will then measure the stereotypes associated with the concept mentioned in the prompt, using the prompt-specific candidate sets.
>
> > **W2** How can OASIS be useful beyond nationality domains? It is not clear in the paper how OASIS can be extended to evaluate other biases in T2I models, such as gender biases, racism, etc.
> >
>
> We have provided examples of OASIS’s usefulness beyond nationality in Table 2 and Sec. 4.1. Given the stereotype candidates for any concept (e.g., nationality, profession, etc.), OASIS can detect and measure the extent of these stereotypes from a set of generated images. These stereotypes could include gender, as shown in Table 1 for different nationalities, and also in Table 2 where we used OASIS to effectively measure and compare the gender stereotypes of SDv2, SDv3, and FLUX for the concept of “doctors” (first column).
>
> > **W3** Overall, this is a solid piece of work; however, a deeper discussion of its limitations and the specific types of stereotypes for which OASIS may or may not be effective would be beneficial.
> >
>
> Thank you for your encouragement. We have mentioned OASIS's limitations in Sec. A.5, and we will discuss any additional limitations raised by the reviewers during the discussion period.

---

### Official Review · Reviewer_n4np · 2024-11-05

**Soundness:** 3
**Presentation:** 3
**Contribution:** 4
**Rating:** 10
**Confidence:** 4

**Summary:**

The proposed work has a novel contribution in an existing research gap. Defining, separating, and quantifying a measure for stereotypes and biases in Image generation models (from text prompt) is a strong case of Ethical Machine Learning. The authors built a foundation of definitions that allowed them to measure estimates of data and image stereotypes. Then they apply that on well-known Text to Image models to uncover the stereotypes risks of applying those models according to the proposed measures/estimates.

**Strengths:**

Addressing a research gap
Clear mythology
Application to existing models

**Weaknesses:**

It would have been nice if the writeup included a detailed case study of qualitative analysis of cases of stereotypes

**Questions:**

- In the abstract the authors said "incorrectly categorizes biases as stereotypes", do you mean the opposite (i.e. "incorrectly categorizes stereotypes as biases")?
- How about using other measures of visual variations of the generated images (e.g. a similarity measure based on feature extraction (e.g. HoG))?

---

> ### Author Response · Authors · 2024-11-17
> **Response to Reviewer n4np**
>
> Thank you for your review and feedback.
>
> > **W** It would have been nice if the writeup included a detailed case study of qualitative analysis of cases of stereotypes
> >
>
> Thank you for your suggestion. We have included a qualitative description of the generated datasets on which we evaluated OASIS in the appendix (Sec. A.7).
>
> > **Q1** In the abstract, the authors said "incorrectly categorizes biases as stereotypes", do you mean the opposite (i.e. "incorrectly categorizes stereotypes as biases")?
> >
>
> We believe our statement in the abstract is not incorrect, and we would like to provide further clarification. Consider this example: A T2I model generates images of doctors, 62% of which are men. This dataset is gender-biased since the number of men and women is not equal (please refer to the common response for a mathematical definition of bias). Prior works use statistical parity to measure stereotypes and conclude that this model contains gender stereotypes, although 62% of the doctors in the US are indeed men [R1], and therefore the T2I model has not over-generalized. Here, they wrongly categorized this bias as a stereotype.
>
> > **Q2** How about using other measures of visual variations of the generated images (e.g. a similarity measure based on feature extraction (e.g. HoG))?
> >
>
> Again, thank you for the suggestion. Although examining the variation in the features can tell us about the visual variety among the generated images, it may not shed light on the cause of this variety. For this reason, we measured attribute-specific variance in WALS.
>
> [R1] Association of American Medical Colleges. (n.d.). *Women are changing the face of medicine in America*. AAMC. Retrieved November 14, 2024, from https://www.aamc.org/news/women-are-changing-face-medicine-america

---

> > ### Comment · Reviewer_n4np · 2024-11-26
> >
> > Ok, sounds good. I have no further comments.

---

### Official Review · Reviewer_EfBv · 2024-11-06

**Soundness:** 4
**Presentation:** 2
**Contribution:** 3
**Rating:** 6
**Confidence:** 3

**Summary:**

In this paper, the authors propose multiple new metrics to understand stereotyping in T2I models.  In particular, they propose a metric for stereotyping that compares the prevalence of a given attribute to a societally-anchored prior probability.  They go on to present numerous other analyses for diversity of images and measuring how differnt stereotypical concepts relate to demographics in latent space.  They show that generally newer models exhibit less stereotyping and that stereotypes arise mostly in the earlier stages of optimization.

**Strengths:**

1. Measuring sterotyping is an important and challenging problem for generative models.

2. The diversity of metrics seem quite interesting to get a more complete picture of how bias and stereotypes arise in these models.

3. The paper is fairly thorough in its testing.

**Weaknesses:**

1. The biggest weakness in my opinion is that the paper is quite hard to follow for many of the metrics. I believe this is due to making the description of the metrics unnecessarily complex, and I would encourage the authors to try to describe all of the concepts more simply.  In addition to making the paper harder to understand, it also obfuscates the insights and contributions.

2. One of the papers claimed main contributions is measuring stereotyping relative to societal priors on the distribution.  However, as the authors state in the appendix, the prior is often unknown and thus falling back to 50% is a reasonable choice.  That said, this seems to limit the contribution over prior work.

3. The paper does not validate their work with human judgements

4. More minor: the paper only presents the evaluations but do not work on mitigations.

5. The intro could be more clear and sensitive in how it characterizes what are stereotypes, why they matter, and how they define which stereotypes are important.

**Questions:**

S3.2 - WALS is quite confusingly described.  In particular (6) doesn't really make sense as an equation.  If E(D) is the SVD that is just estimating the dataset? Do you mean the rank or something else?  Even the motivation doesn't make much sense to me of why this is necessary.

"Existing bias definitions are not applicable for some attributes studied in Tab. 1. For example, in the case of wearing turban, a T2I model needs to depict 50% of the images of Iranian with turban to be unbiased according to Eq. (1), which incorrectly represents Iranian people." - This seems like an odd remark in that this is largely what is done here just conditioned on gender.  What do you see as the primary difference here?

---

> ### Author Response · Authors · 2024-11-17
> **Response to Reviewer EfBv (Part 1)**
>
> Thank you for your detailed feedback. Below are our responses to individual concerns and questions.
>
>
> > W1 The biggest weakness in my opinion is that the paper is quite hard to follow for many of the metrics. I believe this is due to making the description of the metrics unnecessarily complex, and I would encourage the authors to try to describe all of the concepts more simply. In addition to making the paper harder to understand, it also obfuscates the insights and contributions.
> >
>
> We are aware that reading the paper might be challenging due to the terminology necessary to explain various metrics in the paper. To make it easier to follow, we included a one-line description for each metric in Fig. 2, conveying their high-level purposes as given below:
>
> 1. Stereotype Score: Measures the stereotype related to a given attribute in the generated images.
> 2. WALS: Measures the spectral variance of the generated images along the direction of a given attribute.
> 3. StOP: Discovers the attributes the T2I model internally associates with a concept.
> 4. SPI: Measures the emergence of stereotypes during the image generation process.
>
> When each metric is defined, this description is expanded, and an explicit motivation section (in some cases, with examples) is included. If the reviewer has further suggestions to improve the clarity of the writing, we will happily incorporate them into the revised version.
>
>
>
> > W2 One of the paper’s claimed main contributions is measuring stereotyping relative to societal priors on the distribution. However, as the authors state in the appendix, the prior is often unknown and thus falling back to 50% is a reasonable choice. That said, this seems to limit the contribution over prior work.
> >
>
> OASIS can measure the true extent of stereotypes when $P^*(A\mid C, D)$ is known, unlike existing works. Even in the absence of prior knowledge, when we use a uniform prior, OASIS estimates the directional violation of the underlying distribution, which again was ignored in the previous works. For example, we assume that 50% of Mexican men can have a mustache in the concept of Mexican person and mustache. In this case, OASIS will identify the mustache as stereotype only if the ratio of men **with** mustaches exceeds 50%. In contrast, prior works define bias such that the dataset is categorized as biased if *either* the ratio of men **without** mustaches exceeds 50% *or* the ratio of men **with** mustaches exceeds 50%. Please refer to our common response for a comparison of the mathematical definitions of bias and stereotypes.
>
> Furthermore, the mathematical formulation for stereotypes score is only one of our contributions. We also provide methods to understand the origins of these stereotypes in T2I models and measure their spectral variance along the stereotypical attributes, none of which requires $P^*(A\mid C, D)$.
>
> > W3 The paper does not validate their work with human judgements.
> >
>
> As mentioned in the main paper, one of the main goals of the paper is to make the stereotype evaluation process automatic since human judgment can be subjective and expensive. Throughout our work, we used CLIP ViT-G-14 for attribute prediction, similar to (D’Inca et al., 2024; Luccioni et al., 2023). The performance of these VLMs on various attribute prediction tasks is benchmarked. In the paper, we added two analyses in Sec. A.5 on the accuracy of the employed CLIP model in 1) predicting the gender in a human-annotated dataset of generated images of doctors in Tab. 4, and 2) predicting “wearing a hat”, “having a mustache”, “gender”, and “having a beard” in CelebA dataset in Tab. 5. Moreover, we added a qualitative human analysis of the three T2I models on the generated images of four nationalities in Sec. A.7.
>
> > W4 More minor: the paper only presents the evaluations but do not work on mitigations.
> >
>
> We focused on the evaluation of stereotypes in T2I models since it was underexplored among the prior works. Consequently, proposing mitigation techniques was outside the scope of the paper. Nonetheless, we believe that the proposed metrics such as SPI can be used to detect the image generation steps that need interventions to remove stereotypes from the generated images.
>
> > W5  The intro could be more clear and sensitive in how it characterizes what are stereotypes, why they matter, and how they define which stereotypes are important.
> >
>
> We have provided the definition of stereotypes (L52-54), the harms they cause, and why they must be removed (L60-65) in the Introduction. To further emphasize the importance of addressing stereotypes and the harms they can cause, we have added a new discussion in the appendix (Sec. A.6) detailing the harms that stereotypes in generative models can cause, along with examples.

---

> > ### Author Response · Authors · 2024-11-17
> > **Response to Reviewer EfBv (Part 2)**
> >
> > > **Q1** S3.2 - WALS is quite confusingly described. In particular (6) doesn't really make sense as an equation. If E(D) is the SVD that is just estimating the dataset? Do you mean the rank or something else? Even the motivation doesn't make much sense to me of why this is necessary.
> > >
> >
> > We apologize for any confusion that may have led to this misunderstanding. To address this, we would like to clarify the notation here again: $\mathcal{E}_I$ and $\mathcal{E}_T$ are the image and text encoders of the CLIP model, respectively. $\mathcal{D}$ is the dataset containing $N$ images. Therefore, $\mathcal{E}_I(\mathcal{D})$ denotes the representation corresponding to the images in the dataset and is a $N \times m$ matrix where $m$ is the dimensionality of the CLIP’s feature space. This matrix can be decomposed using SVD, as mentioned in the paper. In (6), we have $\delta A = \mathcal{E}_T(d^+) - \mathcal{E}_T(d^-)$ where $d^+$ and $d^-$ are the positive and negative text descriptions, which makes $\delta A$ a vector with a size of $m$ in the feature space of the CLIP model, indicating the direction of change in the attribute $A$.
> >
> > WALS, as mentioned in L225-232 of Sec. 3.2, measures the variance among the images along a given stereotypical attribute. Repeating the example from the main paper, the estimated stereotype score may be 0 for the “male” stereotype for, say, the profession “doctor”. However, a T2I model may achieve this perfect score by repeatedly generating images of the same doctor. WALS aims to quantify this variance.
> >
> > > **Q2** "Existing bias definitions are not applicable for some attributes studied in Tab. 1. For example, in the case of wearing turban, a T2I model needs to depict 50% of the images of Iranian with turban to be unbiased according to Eq. (1), which incorrectly represents Iranian people." - This seems like an odd remark in that this is largely what is done here just conditioned on gender. What do you see as the primary difference here?
> > >
> >
> > This remark shows the difference between the definition of bias and stereotype. For the provided example of Iranian people with turbans, based on the definition of bias that is widely used in the prior works, a T2I model is unbiased if it generates the same number of Iranian people with and without turbans, equivalent to having 50% of the images wearing a turban. However, based on the statistics, only 0.2% of people in Iran wear a turban. Therefore, if more than 0.2% of the generated images contain people wearing a turban, the model “overgeneralizes” turbans for Iranian people, which fits with the definition of stereotype. We have edited the remark in Sec. 4.1 to include these details.
> >
> > Please refer to the common response for a comparison of mathematical definitions of bias and stereotype (”Difference between existing bias definitions and our stereotype definition”).
> >
> > If our responses addressed your initial comments, please consider raising the score.

---

> > ### Comment · Reviewer_EfBv · 2024-12-03
> >
> > Thank you for the clarifications.  To run through them:
> >
> > W1. Thank you for the additional explanations. I still have some confusion eg on WALS which I discuss below.
> >
> > W2. This is an interesting point.  While mathematically minor I can believe that is significant in practice.
> >
> > W3. While it is valuable to evaluate the individual components, I believe this misses some key aspects.  In particular, the results depend on the candidate concept generation and the particular stereotype definition.  It'd be nice if the paper were able to validate both the precision and recall from this concept generation as well as whether humans agree with the bias vs stereotype distinction.
> >
> > Overall, I continue to feel the paper would be a good addition to the ICLR program but with some areas for improvement.

---

> ### Comment · Reviewer_EfBv · 2024-12-02
>
> Thank you for the detailed responses! On the WALS definition in (6), something I am still not understanding is that it sounds like if both $d^+$ and $d^-$ are matrices of size $N \times m$ then $\mathcal{E}_T(d^+)$ and $\mathcal{E}_T(d^-)$, each as an SVD, should each output three matrices: $N \times k$, $k \times k$ and $m \times k$ (matrices $U$, $\Sigma$, $V$ respectively as characterized on https://en.wikipedia.org/wiki/Singular_value_decomposition).  How does this become a vector? I'm guessing this is either a misunderstanding or a notational issue.

---

> ### Author Response · Authors · 2024-12-03
>
> > W1. Thank you for the additional explanations. I still have some confusion eg on WALS which I discuss below. \
> Thank you for the detailed responses! On the WALS definition in (6), something I am still not understanding is that it sounds like if both $d^+$ and $d^−$ are matrices of size $N \times m$ then $\mathcal{E}_T(d^+)$ and $\mathcal{E}_T(d^−)$, each as an SVD, should each output three matrices: $N \times k$, $k \times k$ and $m \times k$ (matrices $U$, $\Sigma$, $V$ respectively as characterized on https://en.wikipedia.org/wiki/Singular_value_decomposition). How does this become a vector? I'm guessing this is either a misunderstanding or a notational issue.
> >
>
> Thank you for your reply. We believe this is a misunderstanding and needs to be clarified. As mentioned in line 187, $d^+$ and $d^−$ are the text descriptions corresponding to the presence and the absence of the stereotypical attribute whose WALS we aim to calculate. Given that $\mathcal{E}_T$ is the text encoder of the CLIP model, $\mathcal{E}_T(d^+)$ and $\mathcal{E}_T(d^-)$ will be vectors in the feature space of a CLIP model. For example, if the employed CLIP model is ViT-L-14, then $\mathcal{E}_T(d^+)$ and $\mathcal{E}_T(d^-)$ are 768-dimensional **vectors**. As a result,  $\delta{A}$ in Eq. (6) represents a vector subtraction:
>
> $
> \delta{A} = \mathcal{E}_T(d^+) - \mathcal{E}_T(d^-).
> $
>
> This results in a vector that indicates the direction of the stereotypical attribute $A$ in the CLIP feature space.
>
> We believe that this misunderstanding comes from a potential confusion between $\mathcal{E}_T(d^+)$ and $\mathcal{E}_I(\mathcal{D})$ where  $\mathcal{E}_T(d^+)$ is a **vector** of the encoded text description and $\mathcal{E}_T(\mathcal{D})$ is a **matrix** of $N$ encoded images.
>
> > W2. This is an interesting point. While mathematically minor I can believe that is significant in practice.
> >
>
> Thank you for your encouraging response.
>
> > W3. While it is valuable to evaluate the individual components, I believe this misses some key aspects. In particular, the results depend on the candidate concept generation and the particular stereotype definition. It'd be nice if the paper were able to validate both the precision and recall from this concept generation as well as whether humans agree with the bias vs stereotype distinction.
> >
>
> We agree that incorporating human judgment in identifying a candidate set for stereotypes related to a concept can enhance the robustness of the results. However, as we clarified in response to reviewer **AMAa**, this does not impact the contributions of OASIS:\
> As outlined in the paper, OASIS provides metrics and analysis for a given set of generated images from a T2I model and a corresponding candidate set of stereotypes. The process of identifying this candidate set is not a contribution of our work and is independent of OASIS, as depicted in Figure 2. In this paper, we adopted a commonly-used approach involving LLMs to derive candidate stereotypes, following prior work in multimodal models (Adila et al., 2024) and generative models (D’Inc et al., 2024). However, OASIS is not restricted to this approach and can work with alternative methods, such as curating the candidate set through domain experts.
>
> To clarify this point further, we will enhance the discussion in the Limitation section (A.5) to explicitly outline these points and mention some possible alternatives for identifying candidate stereotypes beyond LLMs.
>
> > Overall, I continue to feel the paper would be a good addition to the ICLR program but with some areas for improvement.
> >
>
> Thank you again for your comments as they have greatly improved the quality and clarity of our manuscript. We would be happy to address your outstanding concerns.

---

### Author Response · Authors · 2024-11-17
**Common Response (Part 1)**

We thank all the reviewers for their valuable feedback. Our primary goal in this work was to evaluate stereotypes in T2I models so that they may achieve wider use in society, and we are happy that [**EfBv, QudX**] found the problem that we are addressing socially significant and important. As pointed out by [**Hz7R**], we hope that our work leads to more engagement from the research community to build an inclusive AI.

We are also encouraged that the reviewers found our paper well-written [**Hz7R, AMAa**], clear [**Hz7R, QudX**], easy to follow [**Hz7R, AMAa**], interesting and thorough [**EfBv**], with scalable formulation [**AMAa**] that provides valuable insights [**Hz7R, QudX**].

We have incorporated the feedback in the paper and updated it with our changes, which are highlighted in blue. We first summarize the primary concerns of the reviewers and how we addressed them:

- Reviewers **AMAa** and **QudX** asked for more clarity on the definition of the stereotype used in the paper. To address that, we compared bias definitions and our stereotype definition in common response part 2 (”Difference between existing bias definitions and our stereotype definition”), providing mathematical definitions and numerical examples. Moreover, we also justified the use of true distribution of the data in the definition of the Stereotype Score in common response part 2 (”On the ethics of using census data to measure stereotype”).
- Reviewers **EfBv** and **AMAa** asked for further accuracy analysis of the CLIP model used in the paper and validation of its results with human judgments. To address this concern, we added an analysis of the CLIP model's performance on a small manually annotated generated dataset and the CelebA dataset, a large dataset with annotations. This analysis is added to Sec. A.5 of the paper.
- Reviewer **AMAa** requested more samples on SPI. To address this, we generated 100 samples and averaged their SPI for a more robust result. This result further confirmed our observation that stereotypical attributes arose early in the image-generation process. This result is added to Sec. A.3. However, as we mention in our individual response to **AMAa**, the purpose of SPI is to provide a sample-wise stereotype attribute measurement that can be used by the mitigation methods in contrast to other metrics in OASIS that are applied to the distribution of generated data.
- Reviewers **EfBv** and **n4np** requested a qualitative analysis of the generated data. To address that we added a qualitative human analysis of the three T2I models on the generated images of four nationalities in Sec. A.7.

---

> ### Author Response · Authors · 2024-11-17
> **Common Response (Part 2)**
>
> # Difference between existing bias definitions and our stereotype definition
>
> As mentioned in L145-160, stereotypes are over-generalized beliefs for the whole concept, while biases are disparity in the presence of some attribute across different groups. Mathematically, bias for a binary attribute is generally defined as
>
> $|P(A = a | C, \mathcal{D}) - P(A = \neg a | C, \mathcal{D})|$,
>
> while our definition of stereotype score (Eq. 3 in the main paper) captures the violation of the attribute proportion from the true distribution only for the attribute that is candidate to be a stereotype. We denoted that definition as
>
> $\max( 0, P(A = a | C, \mathcal{D}) - P^*(A = a | C))$
>
> where $a$ is the attribute that is prone to become a stereotype.
>
> These definitions demonstrate two critical differences between general bias and stereotype. 1) In the definition of stereotype, the true distribution of the attribute in the data, i.e., $P^*(A = a | C)$, is considered, and 2) the stereotype score, unlike the bias definition, is directional, i.e., only $P(A = a | C, \mathcal{D}) > P^*(A = a | C)$ leads to stereotype and having a lower proportion of the attribute in the data than the true distribution is not considered a stereotype. The consequences of these two points are shown in the results section and in the remark in L348 that highlights the differences between these two.
>
> As an illustration, consider the distributions of gender in the samples generated by three T2I models,  $M_1$, $M_2$, and $M_3$.
>
> | Model name | $P(A=\text{male}\mid C=\text{doctor}, \mathcal{D})$ | $P(A=\text{female}\mid C=\text{doctor}, \mathcal{D})$ | Bias score (existing definition) | Stereotype Score (our definition) |
> | --- | --- | --- | --- | --- |
> | $M_1$ | 0.3 | 0.7 | 0.4 | 0 |
> | $M_2$ | 0.5 | 0.5 | 0 | 0 |
> | $M_3$ | 0.7 | 0.3 | 0.4 | 0.08 |
>
> In this table, the Bias score is calculated as $|P(A = \text{male} | C = \text{doctors}, \mathcal{D}) - P(A = \text{female} | C = \text{doctors}, \mathcal{D})|$ and the Stereotype Score is calculated as $\max( 0, P(A = \text{male} | C=\text{doctors}, \mathcal{D}) - P^*(A = \text{male} | C=\text{doctors}))$.
>
> Here, $M_1$ is a biased but not stereotypical model that produces images of predominantly female doctors. The bias definition says the model is biased, and OASIS gives it a zero stereotype score. This is due to the directionality of the stereotype definition that we follow. $M_2$ is an unbiased model with a zero bias score. OASIS gives it a zero stereotype score despite the model violating the census data distribution. $M_3$ is a biased and stereotypical model that produces images of predominantly male doctors. The model is biased according to the bias score, and OASIS predicts that the model is stereotypical.
>
>
> # On the ethics of using census data to measure stereotype
>
> In the example of generating images of doctors, the notion of stereotype is that “doctors are likely to be men”. Census statistics show that 62% of the doctors in the US are indeed men [R1]. In this context, we say a T2I model contains gender stereotypes only if more than 62% of the images it produces contain men. If over 38% of the images it produces contain women, we **do not** say the model is stereotypical. This directionality in the definition is one of the differences between our stereotype definition and existing bias definitions. As a result, this definition does not require the T2I model to strictly adhere to census data when generating a balanced distribution (50% male and female). Instead, it only prevents the model from producing an over-generalized representation of concepts, such as having more than 62% male doctors.
>
> Moreover, T2I models are intended to be used by the general public for various applications. Therefore, we argue that these models must not be penalized for following the true distribution. For example, consider generating images of “doctors in the US in the 18th century” for a school project. Most doctors were men in the 18th century. Therefore, if a T2I model generates images of predominantly male doctors in this example, we should not flag it as stereotypical.
>
> Please let us know if you have any remaining concerns or questions you would like us to address.
>
> [R1] Association of American Medical Colleges. (n.d.). *Women are changing the face of medicine in America*. AAMC. Retrieved November 14, 2024, from https://www.aamc.org/news/women-are-changing-face-medicine-america

---

### Author Response · Authors · 2024-11-23
**Follow-Up on Rebuttal Discussion**

Dear Reviewers,

Thank you for your valuable feedback on our submission. We have carefully addressed your comments and concerns in our rebuttal and the main paper, which were submitted six days ago.

As the end of the discussion period approaches, we kindly encourage you to share any remaining questions or thoughts to further clarify our responses. If our rebuttal has already addressed your concerns, we would greatly appreciate it if you could consider revisiting or updating your evaluation scores.

We deeply value your time and effort in reviewing our work and look forward to your thoughts.

Thank you!

---

### Meta-Review · Area_Chair_VmYT · 2024-12-29

**Metareview:**

Text-to-image (T2I) models are widespread and tend to exhibit behavior that is negative towards certain communities. This is well documented in the T2I literature. This paper does a nice job separating "stereotyping" from other forms of bias and focusing on measuring it and identifying its origin in T2I models. I agree with the reviewers' assessment of this work and recommend that it be highlighted in the conference with the hope this contributes towards the design of more inclusive T2I models.

**Additional Comments On Reviewer Discussion:**

The authors thoroughly addressed the reviewers' questions and comments in their rebuttal.

---

### Decision · Program_Chairs · 2025-01-22

Accept (Spotlight)